# ROYAL SOCIETY
# OPEN SCIENCE

green chemistry/chemical engineering/ environmental chemistry

$CO_2$ absorption and desorption, heterogeneous catalysis, Eley–Rideal model, Zwitterion mechanism, solid surface reactions

**Authors for correspondence:**
Huancong Shi
e-mail: hcshi@usst.edu.cn
Yuandong Huang
e-mail: huangyd@usst.edu.cn
Linhua Jiang
e-mail: lhjiang@usst.edu.cn

This article has been edited by the Royal Society of Chemistry, including the commissioning, peer review process and editorial aspects up to the point of acceptance.

# Eley–Rideal model of heterogeneous catalytic carbamate formation based on $CO_2$–MEA absorptions with $CaCO_3$, $MgCO_3$ and $BaCO_3$

Huancong Shi[1,2], Min Huang[1], Yuandong Huang[1], Lifeng Cui[1], Linna Zheng[1], Mingqi Cui[1], Linhua Jiang[1], Hussameldin Ibrahim[2] and Paitoon Tontiwachwuthikul[2]

[1]Department of Environmental Science and Engineering, Shanghai Key lab of Modern Optical Systems, University of Shanghai for Science and Technology, Shanghai, 200093, People's Republic of China
[2]Clean Energy Technology Research Institute (CETRI), Faculty of Engineering and Applied Science, University of Regina, 3737 Wascana Parkway, Regina, Saskatchewan S4S 0A2, Canada

HS, 0000-0003-4333-4118

The mechanism was proposed of heterogeneous catalytic $CO_2$ absorptions with primary/secondary amines involving 'catalytic carbamate formation'. Compared with the non-catalytic 'Zwitterion mechanism', this Eley–Rideal model was proposed for $CO_2$ + RR'NH with $MCO_3$ (M = Ca, Mg, and Ba) with four elementary reaction steps: (B1) amine adsorption, (B2) Zwitterion formation, (B3) carbamate formation, and (B4) carbamate desorption. The rate law if determining step of each elementary step was generated based on 'steady-state approximation'. Furthermore, the solid chemicals were characterized by SEM and BET, and this rate model was verified with 39 sets of experimental datasets of catalytic $CO_2$–MEA absorptions with the existence of 0–25 g $CaCO_3$, $MgCO_3$ and $BaCO_3$. The results indicated that the rate-determining step was B1 as amine adsorption onto solid surface, which was pseudo-first-order for MEA. This was the first time that the Eley–Rideal model had been adopted onto the reactions of $CO_2$ + primary/ secondary amines over alkaline earth metal carbonate ($MCO_3$).

## 1. Introduction

The $CO_2$ absorption with alkanolamine solution is an important industrial operation for post-combustion carbon capture (PCCC), and the kinetics of the reactions of $CO_2$–amine solutions are of

considerable interest [1]. The kinetics is one of the key parameters as a database for the design and simulation of the absorption column [2,3]. Therefore, reaction kinetics of $CO_2$−amine solvents has been intensively investigated since 1968 [4]. After Zwitterion mechanism was proposed [4], a large amount of research was conducted for kinetics of primary, secondary and tertiary amines in the period 1979−2015 [5−21]. Since 2012, two reviews [2,3] described summarized statements of reaction kinetics, reaction mechanism, kinetics models and kinetics behaviour of $CO_2$−amine solvents. For the $CO_2$ reactions with tertiary amines, the product is bicarbonate ($HCO_3^-$) and the mechanism is 'base catalysed hydration mechanism' [2]. For the $CO_2$ reactions with primary and secondary amines, the product is carbamate ($RR'N\text{-}COO^-$) and the mechanism is mostly Zwitterion mechanism [4,6], except for Termolecular mechanism under special cases [17]. The focus of this study was based on the Zwitterion mechanism of carbamate formation with heterogeneous catalysis.

Despite intensive kinetic studies of homogeneous $CO_2$ absorptions with amines, there are few studies of heterogeneous catalytic $CO_2$−amine absorptions with the aid of solid chemicals in terms of kinetics analyses, because of limited literature or experimental results [22]. Fortunately, recent studies started to focus on the heterogeneous catalytic $CO_2$−amine absorption. The potential of solid alkaline chemicals in $CO_2$ absorption-MEA was discovered with experimental studies of batch and semi-batch reactions and patented after 2012 [22]. Later on, the experiments were conducted of $CO_2$ absorption of primary and secondary amines MEA [22] and DEA [23] with the aid of $CaCO_3$, $MgCO_3$ and $BaCO_3$. The absorption profiles of $CO_2$−DEA and $CO_2$−MEA of the batch process with stirred reactor and the semi-batch process verified that $MCO_3$ accelerated $CO_2$ absorption by around 10−25% [22,23]. These reactions involved carbamate ($RR'N\text{-}COO^-$) formation from primary amine (MEA) and secondary amine (DEA). In the batch process, the solid chemicals were wrapped and placed into gas−liquid interface and gas was introduced into solid−liquid [23]. In the semi-batch process, they were also installed into the middle of several pieces of inert packing material made of stainless steel [22]. Apart from experimental results, the kinetic models of heterogeneous catalytic $CO_2$ absorption with primary and secondary amines awaited kinetic investigations, such as studies on reaction mechanism, elementary steps, rate-determining steps, reaction orders and rate constant, etc.

Generally, there are three types of kinetic models describing heterogeneous catalysis involving gas−solid reactions: Langmuir−Hinshelwood (LH) model, Eley−Rideal (ER) model, and Mars−van-Krevelen model. However, the Mars−van-Krevelen model involves redox reaction and it is unsuitable for carbamate formation. Similarly, the Langmuir−Hinshelwood model involves the adsorption of both $CO_2$ and MEA onto the solid surface before their reaction. However, it is easier for $CO_2$ to attach to amines which are pre-adsorbed onto solid than adsorbed onto active sites on the surface of solid carbonates. Therefore, the Eley−Rideal model might be suitable for this study.

The Eley−Rideal mechanism describes a reaction between a reactant which has chemisorbed and another one which has not chemisorbed [24]. The defining characteristic of an Eley−Rideal reaction is that one of the reactants is not chemisorbed locally, and hence, not in equilibrium with the surface temperature. This model has been widely adopted into heterogeneous catalysis in gas−solid interface, such areas as $H_2$ adsorption on Ru(001), Fischer−Tropsch Chemistry on Ru(001) [24], synthesis of dimethyl carbonate (DMC) from carbon dioxide ($CO_2$) and methanol (MeOH) over $ZrO_2$−MgO catalyst [25].

Eley−Rideal model has hardly been applied to $CO_2$−MEA interactions before, but the case of $CO_2$ reaction with MeOH to synthesize DMC on a $ZrO_2$−MgO catalyst [25] is similar: $CO_2$ reaction with $RR'NH$ on solid surface. Meanwhile we studied the molecules simulations of $CO_2$−MEA reactions via Zwitterion mechanism [4,6]. This is the first time that we have adopted Eley−Rideal model into the field of $CO_2$−amine reactions involving carbamate formation. Combining theory of both Eley−Rideal model and Zwitterion mechanism, the mimic heterogeneous catalytic reaction process was plotted with elementary reaction steps.

Within this study, we proposed an Eley−Rideal mechanism of heterogeneous catalytic carbamate formation on the surface of solid chemicals. $CaCO_3$, $MgCO_3$ and $BaCO_3$ were selected as a group, for the metals were alkaline earth metals belonging to IIA group in the Periodic Table. It was necessary to generate the elementary reaction steps, rate-determining step, and rate equations combined with experimental verifications here. We completed several tasks in this work: (1) propose a mechanism of heterogeneous catalytic carbamate formation on the solid surface of $MCO_3$ with Eley−Rideal model; (2) characterize catalyst surface with SEM and BET methods; (3) develop four elementary steps with rate-determining steps (RDS) of each sub-case. We developed the rate law of heterogeneous catalytic reactions based on derivation of different rate-limiting steps or rate-determining steps [26]. (4) Verify that the RDS is the first step as 'amine adsorption' onto solid surface for the case of $CO_2$−MEA absorption with experimental datasets. The suitable rate law if rate limiting/determining [26] was verified by rate models verification $F(X_A) = Kt$ based on experimental data [27]. (5) Compare the rate

equations with the non-catalytic one and estimate the enhancement of solid chemicals. This Eley–Rideal model was suitable for describing catalytic $CO_2$ absorption with primary/secondary amines, which was quite useful for the kinetic studies of heterogeneous catalysis in the field.

# 2. Theory: mechanisms of non-catalytic and catalytic carbamate formation

## 2.1. Zwitterion mechanisms: the role of [OH⁻] and solid alkaline ($W_B$)

The main reactions are listed below of $CO_2$ reaction with primary/secondary amines in aqueous solutions, firstly [2,6]. Blauwhoff et al. [6] have already generated the main equations after conducting the kinetics of $CO_2$–amine in aqueous solutions.

$$\textbf{Carbamate formation} \quad CO_2 + 2\,R_1R_2NH \rightleftharpoons R_1R_2N\text{-}COO^- + R_1R_2NH^+ \tag{2.1) [2,6]}$$

$$\textbf{Bicarbonate formation} \quad CO_2 + OH^- \rightleftharpoons HCO_3^-. \tag{2.2) [2,6]}$$

Then, the overall reaction rate consists of two parts below:

$$r_{ov} = r_{CO_2-R_1R_2NH} + r^*_{OH^-} = k_{ov}[CO_2] = k_{app}[\text{amine}][CO_2] + k^*_{OH^-}[OH^-][CO_2]. \tag{2.3) [2,6]}$$

Furthermore, Blauwhoff et al. [6] developed the rate constants in detail. The overall rate constant $k_{ov}$ covers the contributions of both reactions and can be written as below:

$$k_{ov} = k^*_{OH^-}[OH^-] + \frac{k_{2,R_1R_2NH}[R_1R_2NH][CO_2]}{1 + (k_{-1})\,/(k_{2,R_1R_2NH}[R_1R_2NH] + k_{OH^-}[OH^-] + k_{H_2O}[H_2O])}. \tag{2.4) [6]}$$

The bicarbonate formation was not dominant if compared with carbamate formation for condensed amine solutions [2,6]. The rate equation $r^*_{OH^-}$ and constant $k^*_{OH^-}$ has already been developed [2,21].

$$r^*_{OH^-} = k^*_{OH^-}[OH^-][CO_2] \tag{2.5) [2,6,21}$$

and

$$k^*_{OH^-} = 8322 \text{ m}^3/\text{kmol at 298 K}. \tag{2.6) [21]}$$

The focus of this study was on 'carbamate formation', in terms of reaction rate $r_{CO_2-R_1R_2NH}$ and rate constant $k_{app}$. They were much bigger than $r^*_{OH^-}$ and $k^*_{OH^-}$ [2]. Based on the recent review [2], both Zwitterion mechanism [4,6] and Termolecular mechanism [17,21] are suitable for $CO_2$–MEA interactions, with rate equations of equations (2.4) and (2.7–2.9) [6,17,21]. For MEA solutions, Zwitterion mechanism is suitable for most cases, for it is the first order for [MEA] and [$CO_2$] [2,6]. Termolecular mechanism is more suitable for MEA at high loadings [17],

$$r^Z_{CO_2-R_1R_2NH} = \frac{k^Z_{2,R_1R_2NH}[R_1R_2NH][CO_2]}{1 + (k^Z_{-1})/(k^Z_{2,R_1R_2NH}[R_1R_2NH] + k_{OH^-}[OH^-] + k_{H_2O}[H_2O])}$$
$$= k^Z_{2,R_1R_2NH}[R_1R_2NH][CO_2] \tag{2.4) [6]}$$

($k^Z_{-1}$ is negligible for MEA [6]).

$$k^Z_{2,\,MEA}(\text{m}^3/\text{kmol}\cdot\text{s}) = 4.4\ \times 10^{11}\ \exp\!\left(\frac{-5400}{T}\right) \tag{2.7) [6]}$$

$$r^T_{CO_2-R_1R_2NH} = (k^T_{R_1R_2NH}[R_1R_2NH] + k_{OH^-}[OH^-] + k_{H_2O}[H_2O])[R_1R_2NH][CO_2] \tag{2.8) [17]}$$

$$k^T_{MEA}(\text{m}^6/\text{kmol}^2\cdot\text{s}) = 4.6\ \times 10^9\ \exp\!\left(\frac{-4412}{T}\right). \tag{2.9) [21]}$$

From Zwitterion mechanism, the potential of [OH⁻] was acknowledged [4]. Several 'solid alkaline earth metal carbonates' were selected as heterogeneous catalysts [23]. Catalytic $CO_2$–DEA absorptions were conducted with the aid of $CaCO_3$ and $MgCO_3$ and verified catalysis [23]. Such solid alkalis acted as Lewis base to enhance [OH⁻] catalysed carbamate formation. Roles of both alkalis are grouped in table 1.

**Table 1.** Comparison of non-catalytic and heterogeneous catalytic $CO_2$ absorption.

| reactions | catalysis | Caplow 1968 [4] homogeneous[a] | this work heterogeneous[a] |
|---|---|---|---|
| carbamate formation | catalytic phases | liquid | solid – liquid |
| ($CO_2$ + RR′NH) | active compounds[b] | Brønsted base | solid alkaline carbonate |
| | | $OH^-$ | $CaCO_3$, $BaCO_3$ $MgCO_3$, etc. |
| | mechanism | Zwitterion | Eley – Rideal |

[a]The $CO_2$ absorption reaction is no doubt gas – liquid heterogeneous; but the *catalysis* where the catalytic reaction takes place can be either homogeneous (hydroxide ion) or heterogeneous (solid alkaline catalysts).
[b]For non-catalytic carbamate formation, $[OH^-]$ is responsible for catalytic pathway [4].

From Caplow, $[OH^-]$ facilitates carbamate formation via a catalytic pathway [4]. This idea has been verified with 15–20 primary and secondary amines [4]. From equation (2.3), the first term is via an uncatalysed pathway, while the second term is via a hydroxide-catalysed pathway [4]. The $[OH^-]$ anion in liquid enhances $CO_2$ aminolysis. The second term $k_{OH^-}^* [OH^-]$ of equation (2.3) could affect both $k_{OH^-}^*$ and $k_{app}$ [6]. For the first term, aminolysis of $CO_2$ are converted to two products: carbamate ($R_2N-CO_2^-$) and bicarbonate ($HCO_3^-$). The ratio of carbamate/bicarbonate was presented as equation (2.10) [5]:

$$r_{ov} = k_{ov}[CO_2] = r_{CO_2-R_1R_2NH}^Z + r_{OH^-}^*$$
$$= k_{app}[amine][CO_2] + k_{OH^-}^*[OH^-][CO_2] \quad\quad (2.3) \; [2,6]$$

$$\frac{carbamate}{(bi)carbonate} = \frac{k_{amine}(amine) + k'_{amine}(amine)(OH)}{k_{OH}(OH)} \quad\quad (2.10) \quad [5]$$

$$r = k_{amine}(R_2NH)(CO_2); \; k_{amine} = \frac{k_1 k_2}{k_{-1}K + k_2} \quad\quad (2.11) \quad [4]$$

The rate-limiting step of Zwitterion mechanism is determined by the relative size of $k_2$ and $k_{-1} \cdot K$ in equation (2.11) [4]. For amines like MEA and DEA, the reaction of carbamate formation is fast with $k_{-1} \cdot K < k_2$. The dissociation of the complex to carbamate and hydronium ion is faster than the loss of $CO_2$. For some other amines, the reaction is relatively slow with $k_{-1} \cdot K > k_2$. The rate-limiting step is C-N bond formation or cleavage, and the $CO_2$ expulsion out of Zwitterion is easier than proton loss [4].

For heterogeneous catalysts, solid alkaline significantly enhances the catalysis over trace $[OH^-]$. There are three advantages of solid alkaline over $[OH^-]$ ions in liquid. Firstly, the homogeneous catalysis is relatively weak because the concentration of $[OH^-]$ is negligible and constrained by the vapour–liquid equilibrium of amine–$CO_2$–$H_2O$ system [3]. The solid catalysts are abundant and vary with different masses. Secondly, the $[OH^-]$ is detrimental to the $CO_2$ desorption process, which can hardly be separated out of the solution, but the insoluble solid alkaline chemicals could be separated out of the liquid phase. Thirdly, different types and masses of solid catalysts can be placed into the layers between structured packing materials in a packing column of $CO_2$ absorber [22], which is highly tunable. From experimental procedures [22,23], $CO_2$ absorption with bubbling was quite effective when solid chemicals were suspended at gas–liquid interface, indicating that the heterogeneous catalytic $CO_2$ absorption was more likely to occur on the surface.

## 2.2 Proposed mechanism of heterogeneous catalysis: Eley – Rideal model

As mentioned, both Eley–Rideal model and Langmuir–Hinshelwood model mechanisms were applicable in the field of heterogeneous catalytic reactions on gas–solid interface [26]. After detailed investigation and analysis, Eley–Rideal model could be more suitable for carbamate formation reaction because of the acid–base nature of $CO_2$ and $RNH_2$. $CaCO_3$, $MgCO_3$ and $BaCO_3$ contained abundant basic active sites on the surface, and they were pre-absorbed with MEA molecules from experimental procedures. From figure 1, the $CO_2$ molecules reacted with MEA that was instantaneously adsorbed on the solid surface when approaching the surface. $CO_2$ was unlikely to be

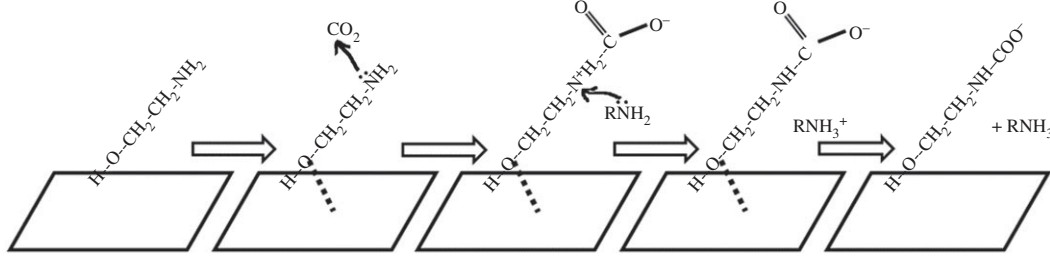

**Figure 1.** Mechanisms of Catalytic carbamate formation based on Eley–Rideal model.

directly adsorbed onto the solid surface. If $CO_2$ was adsorbed onto the solid surface without collision of any amine molecules, the carbamate formation reaction did not occur at all. In the Eley–Rideal mechanism, a gas-phase reagent such as $CO_2$ directly reacted with an adsorbed species (adspecies) RR′NH, and the product carbamate either desorbed or remained adsorbed on the surface depending on the exothermic reaction [24]. Mostly, the product desorbed the surface due to the heat release [24].

The heterogeneous catalytic carbamate formation is proposed in figure 1. The molecular interaction of $CO_2$, MEA and active sites '*' were similar to the proposed mechanism of catalytic $CO_2 + MeOH$ over $ZrO_2$–MgO catalyst via the ER model [25]. Figure 1 illustrates the molecular interactions on the solid surface. A large number of MEA molecules had been pre-attached to the active sites on the solid surface according to experimental procedures. When the gas bubbles (containing pure $CO_2$) hit the catalyst surface, $CO_2$ molecule was transferred onto the amine adsorbed on active sites, where the heterogeneous catalytic reactions took place. The solid surface area was much larger than the gas–liquid interface, and it facilitated $CO_2$ absorptions via enhanced mass transfer. In short, figure 1 is the Eley–Rideal model of catalytic $CO_2 + MEA$ with gas–solid interaction, except that the solid surface was covered by amine solvents with MEA and $H_2O$ molecules.

The apparent rate law was developed [26] with four elementary reaction steps: (B1–B4) in table 2, where 'B' represents 'base'. Amine adsorption was the start-up (B1). $CO_2$ reacted with MEA to generate Zwitterion (B2). Another water/base accepted the protons transferred from the Zwitterion, and carbamate was formed with heat release (B3). The carbamate desorption with completed reaction (B4). In short, B1 was amine chemisorption, and B2 was $CO_2$ aminolysis/Zwitterion formation. B3 was proton detachment/carbamate formation, and B4 was carbamate desorption. The amine adsorption (B1) involved the mass transfer of MEA from bulk of liquid onto solid surface. According to literature, Zwitterion formation without solid catalysts required activation energy Ea of 9.6–10.2 kcal mol$^{-1}$ (40.2–42.7 kJ mol$^{-1}$) under non-catalytic reaction with simulation [28] and about 40 kJ mol$^{-1}$ with experiments [19]. The exact activation energy of B2 was not tested, but it could be calculated with simulation or tested with further kinetic analysis with the equation of lnk = Ea/RT. The carbamate formation (B3) is exothermic [2]. The product, carbamate, does not fit to the increased surface temperature. It desorbs with high translational and internal energy via B4 [26], depending on the exothermic reaction of B3.

These elementary steps were suitable for both primary and secondary amines (MEA, DEA, DIPA, MMEA etc., with unified form of RR′NH) with 'carbamate (RR′N-COO$^{-}$)' as product. However, they were unsuitable for tertiary amines ($R_3N$). The $CO_2$-$R_3N$ reactions involved different reaction schemes, mechanisms and products of bicarbonate ($HCO_3^-$) for $CO_2$ reaction with tertiary amines ($R_1R_2R_3N$) so that B1–B4 were unsuitable for it. However, different primary and secondary amines have different values of $k_{-1} \cdot K$ and $k_2$ in equation (2.7) [4], so that each amine had its own rate-determining step (RDS) among B1–B4. The development of apparent rate law derivations based on these elementary steps was similar to the electronic supplementary material of catalytic $CO_2$ with MeOH for the synthesis of DMC [26]. Compared with non-catalytic Zwitterion mechanism, both mechanisms contained the elementary reaction steps of carbamate formation and Zwitterion de-protonation under different phases. The Eley–Rideal model contained adsorption (B1) and desorption of amine (B4) on the solid surface.

## 2.3 The rate equations of four elementary steps and the suitable RDS of MEA solvents

The rate law if rate determining of elementary reaction steps B1–B4 was developed with details in Support Information A in the electronic supplementary material. This methodology was the same as

**Table 2.** The elementary reaction steps and rate law if rate determining of figure 1.

| no. | elementary reaction steps of catalytic carbamate formation Eley–Rideal model[a] | rate constant |
|---|---|---|
| B1 | $RNH_2 + (*) \rightleftarrows RNH_{2(*)}$ | $k_1, k_{-1}$ |
| B2 | $CO_{2(g)} + RNH_{2(*)} \rightleftarrows RNH_2^+\text{-}COO^-_{(*)(Zwitterion)}$ | $k_2, k_{-2}$ |
| B3 | $H_2O + RNH_2^+\text{-}COO^-_{(*)} \rightarrow RNH\text{-}COO^-_{(*)(Carbamate)} + H_3O^+$ | $k_3, k_{-3}$ |
| B4 | $RNH\text{-}COO^-_{(*)} \rightleftarrows RNH\text{-}COO^-_{(Carbamate)} + (*)$ | $K_4, k_{-4}$ |
| RDS | rate law if rate determining, with full and simplified formats | |

B1
$$r_1 = K_B \left\{ \frac{[A] - K_A(([C][H^+])/p_B)}{1 + K_4[C] + K_3K_4[C][H^+] + ((K_2K_3K_4[C][H^+])/p_B)} \right\}$$

$$r_1 = K_B \left\{ \frac{[A]}{1 + K_4[C]} \right\} \text{ (simple)}$$

B2
$$r_2 = K_B \left\{ \frac{[A]p_B - K_1K_2K_3K_4[C][H^+]}{K_1 + K_1K_4[C] + K_1K_3K_4[C][H^+] + [A]} \right\}$$

$$r_2 = K_B \left\{ \frac{[A]p_B}{K_1 + K_1K_4[C] + [A]} \right\} \text{ (simple)}$$

B3
$$r_3 = K_B \left\{ \frac{K_1K_2[A]p_B - K_3K_4[C][H^+]}{1 + K_4[C] + K_1K_2[A]p_B + ([A]/K_1)} \right\}$$

$$r_3 = K_B \left\{ \frac{K_1K_2[A]p_B}{1 + K_4[C] + K_1K_2[A]p_B + ([A]/K_1)} \right\} \text{ (simple)}$$

B4
$$r_4 = K_B \left\{ \frac{((K_1K_2)/K_3)[A]p_B - K_3K_4[C][H^+]}{[H^+] + ((K_1K_2[A]p_B)/K_3) + K_1K_2[A][H^+]p_B + ([A]/K_1)[H^+]} \right\}$$

$$r_4 = K_B \left\{ \frac{((K_1K_2)/K_3)[A]p_B}{((K_1K_2[A]p_B)/K_3)} \right\} = K_B \text{ (simple)}$$

[a]Refer to electronic supplementary material for the details of rate equations development.

the 'derivation of apparent rate laws' based on elementary steps + rate-limiting steps of the 'CO$_2$ + MeOH reactions' via Eley–Rideal model [25], which was introduced in the electronic supplementary material [26]. It was originated from the steady-state approximation, a classical method in the kinetics of reaction rates [25–27]. The steady-state approximation involved setting the rate of change of a reaction intermediate in a reaction mechanism equal to zero, and then kinetic equations could be simplified. For instance, if B1 was assumed to be the rate-determining step, the other three steps were regarded as steady-state, and the rate of the formation of the intermediate equalled the rate of its destruction, so that the overall apparent rates $r_2$ to $r_4$ were set to 0 [26]. Therefore, $r_1$ was generated with exact equation. The same works were developed on $r_2$, $r_3$ and $r_4$, where B2, B3, and B4 were assumed to be rate limiting. This overall rate law if rate determining was developed in table 2 with full and simplified formats. The simplification was conducted based on chemical reactions and liquid conditions. Within basic amine solution, [H$^+$] was negligible ([H$^+$] $< 10^{-8} \approx 0$). The $P_{CO2}$ was 1 atm.

Finally, we started to evaluate the exact rate-determining step for MEA as the special case. Although B1–B4 were equally possible to be RDS for various primary and secondary amines, the instantaneous reaction kinetics of CO$_2$–MEA should exclude B3 and B4. Neither of them was likely to be the rate-determining (slowest) step. For B3 (proton transfer), the surrounding was basic with massive MEA. Zwitterion deprotonation is instantaneous $k_{-1} \cdot K \ll k_2$ for MEA (much easier to lose proton than N-C bond cleavage) [4]. Moreover, carbamate formation was completed under B4 (desorption). Herein, the possible rate-determining step could be either B1 or B2 for MEA, indicating that the heterogeneous catalytic carbamate formation was either amine adsorption controlled (B1) or reaction controlled (B2). The step of B1 took time, and B2 had an energy barrier (Ea) of N-C bond formation.

# 3. Experimental process and rate model verification

The catalytic rate model was verified with (1) solid catalyst characterization (2) experimental data of $(\alpha, t)$ with calculated dataset of $(X_A, t)$, and (3) integrated method, representing $F(X_A)$ versus time under different cases and sub-cases.

## 3.1 Catalyst characterization and catalytic $CO_2$ absorption with MEA

The characterization of solid $CaCO_3$, $MgCO_3$ and $BaCO_3$ was conducted with the scanning electron microscope (SEM) (FEI XL-30). $CaCO_3$ and $MgCO_3$ were tested by Brunauer–Emmett–Teller (BET) in surface area, pore size and surface area of solids. The SEM was operated at an acceleration voltage of 25 kV. The BET was measured at 77 K on a BeiShiDe 3H-2000PS4 apparatus. Since they were solid chemicals that were commercially available, there was no need to conduct XRD or XPS analyses. The BET of $BaCO_3$ were conducted [29] with a BET ASAP 2020 from Micromeritics (Georgia, USA). It was degassed at 150°C for 5 h. Barrett–Joyner–Halenda (BJH) method was employed to calculate surface area, pore volume and pore size from absorption/desorption isotherms [29].

 The $CO_2$ absorption process was similar to that of other works [23]. A set of stirred-cell reactor was built with the suspension of pelletized chemicals for the experiments, and the internal diameter of the reactor was 8.4 cm, with a constant interfacial area of 55.4 cm$^2$. The solids were wrapped into two balls and suspended onto the gas–liquid interface. The intersection area of two solid balls was about 9.8 cm$^2$ (2.5 cm in diameter for each). The amine solvents were pre-filled into the reactor, with part of the MEA solvents pre-adsorbed onto the solid surface. The reactor was placed in a cool water bath with a magnetic stirrer. The $CO_2$ flow rate was in the range of $1.0–1.5\,l\,min^{-1}$. A thermometer was placed inside the reactor to detect the temperature. Pure $CO_2$ was introduced to a water-scrubbing process and then flowed into the batch reactor containing amine solvents with bubbles.

 After being introduced into amine solvents, the $CO_2$ reacted with MEA solvent via both non-catalytic and catalytic reaction pathways. For the non-catalytic pathway, the $CO_2$ directly reacted with MEA solvent, and the kinetics had been intensively studied [2]. For the catalytic pathway, the $CO_2$ reacted with the MEA molecules pre-adsorbed onto the solid surface, which was the focus of this study.

## 3.2 The data of $X_A$ versus time, for $CO_2$–MEA at a range of $X_A < 0.80$

Afterwards, full sets of $CO_2$–MEA absorption experiments were conducted with $CaCO_3$, $MgCO_3$ and $BaCO_3$ to provide database of $(X_A, t)$ from the experimental data of $(\alpha, t)$. The mass of chemicals was selected as 5, 10, 15 and 20 g (25 g for $BaCO_3$). Amine concentrations were 1, 3 and 5 $mol\,l^{-1}$, respectively. The absorption profiles of $CO_2$ loading $(\alpha)$ versus time had been completed elsewhere. The $CO_2$ loading $(\alpha)$ of the amine solutions was tested with Chittick apparatus, and the AAD% of the experimental tests was 2.5% [23]. The conversion of amines $X_A$ was calculated from $CO_2$ loading $(\alpha)$ with equations below.

 The reactions of carbamate formation (1) and bicarbonate formation (2) were listed:

$CO_2 + 2MEA \rightarrow NH_2\text{-}CH_2\text{-}CH_2\text{-}COO^- + MEAH^+$ (1) $(\alpha < 0.40; X_A < 0.80)$
$CO_2 + MEA + H_2O \rightarrow HCO_3^- + MEAH^+$ (2) $(\alpha > 0.40; X_A > 0.80)$

The relationship of concentration of free amine [amine], concentration of product carbamate [carbamate] and conversion $X_A$ from $CO_2$ loading were listed $(\alpha)$ below:

$$X_A = 2 \times \alpha \qquad (3.1) \quad (\alpha < 0.40; X_A < 0.80)$$

$$[amine] = C_{A0}(1 - 2 \times \alpha) = C_{A0}(1 - X_A) \qquad (3.2)$$

$$[C_{carbamate}] = C_{A0} \times \alpha = \tfrac{1}{2}C_{A0}X_A \qquad (3.3)$$

 Equations (3.1)–(3.3) are accurate for equation (2.1) at $CO_2$ loading less than 0.40 mol mol$^{-1}$ $(X_A < 0.80)$, based on the ion speciation plots of MEA–$CO_2$–$H_2O$ systems [30]. The product for $CO_2$ reaction with MEA is carbamate. The bicarbonate [$HCO_3^-$] started to be detectable when $\alpha > 0.40$ mol mol$^{-1}$ [30] and both reactions occur in equations (2.1) and (2.2) and the stoichiometric ratio of $CO_2$–MEA is not surely 1:2 [30]. Therefore, the data were adopted as $(X_A, t)$ based on equations (3.1)–(3.3), where $0 < X_A < 0.80$. Twelve sets of data of both $CaCO_3$ and $MgCO_3$ were adopted, and 15 sets of data of $BaCO_3$ were adopted. The data of $[X_A, t]$ are shown in electronic supplementary material,

table SA.1–SA.3, along with figures in §4.2. Each curve represents $CO_2$ absorption with different amine concentrations and types of catalysts with mass ($C_{A0}$, $MCO_3$, W).

## 3.3 The integral method of analysis with $F(X_A) = kt$ for RDS (B1, B2)

There was few precedent researches of rate model verification of the developed equations of B1–B4 in table 2 and the parameters of $K_1$–$Kx$ were hard to calculate, so that the rate equations needed to be verified from the definition of reaction rates $r_A = -((dC_A)/(dt))$ as the fundamental of reaction kinetics [27]. With database of ($X_A$, t) in electronic supplementary material, table SA.1–SA.3, 'integral method of analysis' was adopted for rate equation validation [7], which was a fundamental methodology of chemical reaction engineering [27].

The goal was to establish the direct correlation of '$r = C_{A0}((dX_A)/(dt))$ (3.4)' with its integrated format of '$F(X_A) = Kt$'. We developed several proper integrated equations of $F(X_A) = Kt$ with the combination of the definition of reaction rate of equation (3.4) and the specific format of reaction rate ($r$) of equations (3.5)–(3.7) in order to verify B1 or B2 as the rate-determining step. The $[CO_2]$ was not included, for it had already been verified as the first order for absorption [2].

The rate equation of $CO_2$ absorption was equation (3.4) by amine concentration ($C_A$) or conversion ($X_A$) at the right side,

$$r_A = -\frac{dC_A}{dt} = -\frac{dC_{A0}(1 - X_A)}{dt} = \frac{C_{A0}dX_A}{dt}. \tag{3.4}$$

For the left side, $r_A = f(X_A)$ was adopted by either B1 or B2 in table 2. The integrated rate equations were carried out with equations (3.5)–(3.7) under different sub-cases with brief analyses in Appendices. There was only one set of dataset of ($X_A$, t) obtained from experiments. However, there were several different formats of $F(X_A)$ equations (3.5)–(3.7) based on different rate equations ($r$) in table 2. Consequently, there was one accurate model fitting data ($X_A$, t) among equations (3.5)–(3.7), different sub-cases included

$$RDS = B1 : r_1 = K_B \left\{ \frac{[A]}{1 + K_4[C]} \right\}$$

$$\ln \frac{1}{1 - X_A} = K_B\, t \quad r_1 = K_B[A] \tag{3.5} \qquad \frac{\theta_{carbamate}}{\theta_{empty}} \ll 1$$

$$\ln \frac{1}{1 - X_A} - X_A = K'\, t \quad r_1 = K_B \frac{[A]}{K_4[C]} \tag{3.6} \qquad \frac{\theta_{carbamate}}{\theta_{empty}} \gg 1$$

$$RDS = B2 : r_2 = K_B \left\{ \frac{[A]p_B}{K_1 + K_1K_4[C] + [A]} \right\}$$

$$k_a \left\{ \ln \frac{1}{1 - X_A} - X_A \right\} + X_A = K''\, t. \tag{3.7}$$

$k_a = 1/2\, K_1K_4 = 0.05$, 0.005, and 0 for sub-cases.

For equation (3.7), $k_a$ reflected the ratio of $\theta_{carbamate}/n_{sites}$. To verify the rate models of RDS = B2, $k_a$ was selected as 0.05, 0.025 and 0.005, representing that 10%, 5% and 1% of the total active sites were covered with carbamate. The extreme condition was also tested, where $k_a = 0$ (similar to 0th order).

Each sub-case of equations (3.5)–(3.7) was verified by 39 sets of experimental data of ($X_A$, t). Each set of ($X_A$, t) with its specific parameter (concentration, catalyst, mass, i.e. 1.0 mol l$^{-1}$, $CaCO_3$, 5 g) generated different sets of ($F(X_A)$, t) based on different RDS of equations (3.5)–(3.7). Different curves of ($F(X_A)$, t) reflected different rate equations ($r$). For each set of data ($F(X_A)$, t), there were five points with different conversion of $X_A = 0.0$, 0.4, 0.5, 0.6, 0.7 and 0.8. These curves of ($F(X_A)$, t) could be either linear or curvy.

These integrated rate equations (3.5)–(3.7) were developed with a general format of '$F(X_A) = Kt$', which was a **linear** equation of '$Y = kX$'. If the curves of ($F(X_A)$, t) were linear and straight, it meant ($F(X_A)$, t) was quite fitting for '$F(X_A) = Kt$' of equations (3.5)–(3.7), and its related specific rate equation ($r$) was fitting for the mechanism of B1 or B2. The higher $R^2$ of the lines of '$F(X_A) = Kt$' of equations (3.5)–(3.7) indicated the better fitting of the dataset of '$r = C_{A0}((dX_A)/(dt))$' responded to experimental dataset ($X_A$, t). Therefore, we selected the highest standard $R^2 > 0.99$ as the criteria. After repeated verifications of 39 sets of experimental data ($X_A$, t) with linear regressions, the set containing most lines of $R^2 > 0.99$ of equations (3.5)–(3.7) was verified as the highly accurate rate model.

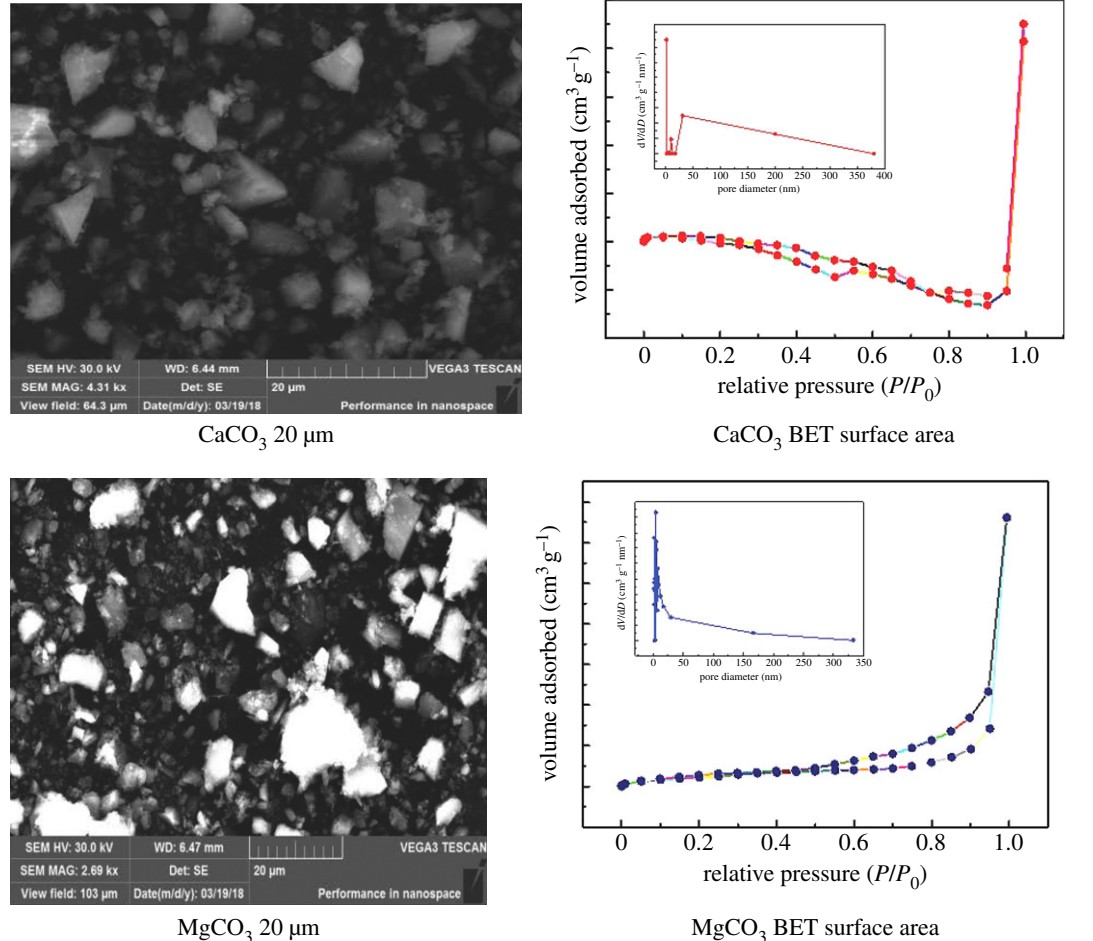

**Figure 2.** SEM and BET of CaCO$_3$, MgCO$_3$.

# 4. Results and discussion

## 4.1 Catalyst characterization

The results include four parts: (1) SEM and BET of solid chemicals; (2) the absorption profiles of $X_A \sim t$ for 39 curves of $CO_2$–MEA absorption with the existence of solid chemicals; (3) RDS verifications of B1 or B2 with experimental data; (4) kinetic rate equations for both heterogeneous catalytic and non-catalytic reactions.

Figures 2 and 3 represent SEM of CaCO$_3$, MgCO$_3$ and BaCO$_3$ at 20 μm, along with BET for CaCO$_3$ and MgCO$_3$. From BET, the surface areas were 0.428 m$^2$ g$^{-1}$ for CaCO$_3$ and 9.498 m$^2$ g$^{-1}$ for MgCO$_3$ from this work. BaCO$_3$ is reported by others as 4.66 m$^2$ g$^{-1}$ for surface area, pore volume of 0.008 cm$^3$ g$^{-1}$ and pore size of 6.46 nm [29]. These surface areas were much larger than gas–liquid interfacial area (55.4 cm$^2$) of the reactor [23]. The pore diameters were 31.3 nm (CaCO$_3$), 4.31 nm (MgCO$_3$) from this study and 6.46 nm (BaCO$_3$) from other work [29], which facilitated the external mass transfer of MEA molecules onto solid surface.

## 4.2 The experimental results of absorption profile ($X_A \sim t$)

Figures 4–12 represent the conversion $X_A$ versus time of 1, 3 and 5 M MEA with the existence of CaCO$_3$, MgCO$_3$ and BaCO$_3$. The experimental data are categorized in electronic supplementary material, table SA.1–SA.3. Based on the figures, it was quite clear that the slopes of curves at conversion range of 0.0–0.80 were quite different from that at $X_A > 0.80$. The absorption curves with catalysts were steeper than the absorption curves under non-catalytic conditions. These figures verified that these solid chemicals were effective in the acceleration of CO$_2$ absorption. The absorption rate increased

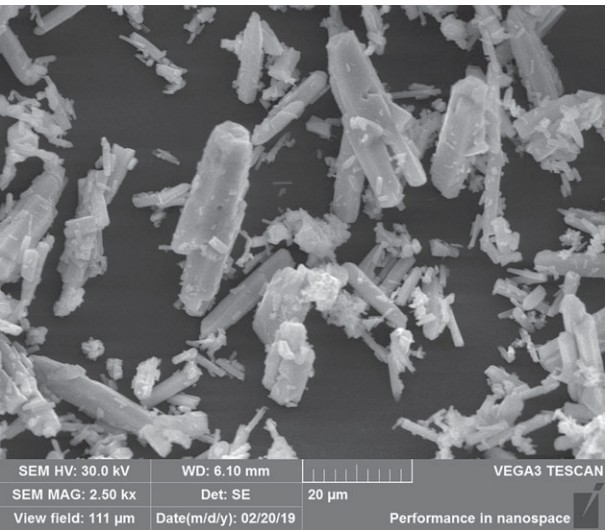

**Figure 3.** SEM of $BaCO_3$.

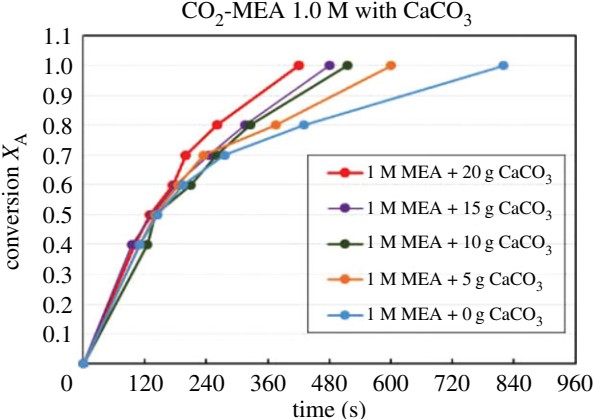

**Figure 4.** Catalytic $CO_2$ absorption of 1.0 M MEA solvents with $CaCO_3$ conversion ($X_A$) versus contact time ($t$).

with increased amounts of solid chemicals until it reached the maximum value of 15 g $MgCO_3$, 20 g $CaCO_3$ and 25 g $BaCO_3$. The effects of catalysts at condensed concentration of 3 M and 5 M were stronger than the effects at dilute concentration of 1 M.

## 4.3 The rate-determining step verification of B1 and B2 based on $F(X_A) = Kt$

The verification of equations (3.5)–(3.7) from $F(X_A) = Kt$ are categorized in table 3. Equations (3.6) and (3.7) did not fit the experimental data accurately. However, equation (3.5) was verified as highly accurate, with 36 of 39 lines of $R^2 > 0.99$, indicating $r_1 = K_B[A]$. Figures 13–21 provide linear regressions for 1, 3, and 5 M MEA with $CaCO_3$, $MgCO_3$ and $BaCO_3$, and exhibit high accuracy. The results indicated that the rate-determining step was B1 (amine adsorption) and ($\theta_{carbamate}/n_{sites}$) $\ll 1$. The solid surface had abundant empty active sites with very little carbamate adsorbed.

The overall $CO_2$ absorption process with the existence of solid alkaline was briefly explained from B1 to B4 in table 2. The amine adsorbed onto the surface from liquid phase firstly, with relatively slow rate (B1). Then $CO_2$ reacted with MEA ($RNH_2$) with N–C bond formation (B2). The rate was instantaneous and enhanced with heterogeneous catalysis. The Zwitterion released proton to $H_2O$ or other base to generate carbamate ($RNH\text{-}COO^-$) and the exothermic reaction released heat (B3). The released heat facilitated diffusion and drove desorption of carbamate back to the aqueous phase (B4). The carbamate finally desorbed the surface due to the exothermic reaction [24]. Mostly, the product desorbed the solid surface due to the heat release, and there was little carbamate remaining.

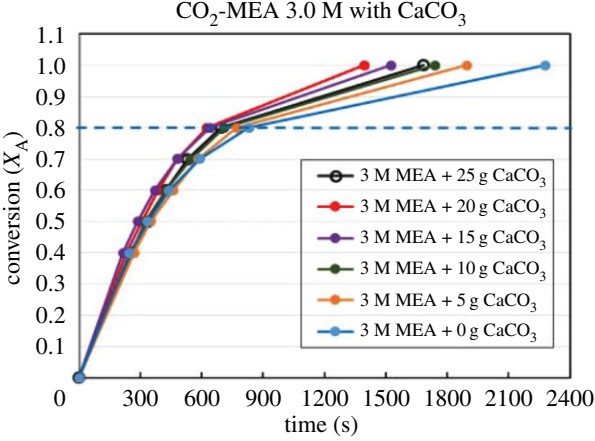

**Figure 5.** Catalytic $CO_2$ absorption of 3.0 M MEA solvents with $CaCO_3$ conversion ($X_A$) versus contact time ($t$).

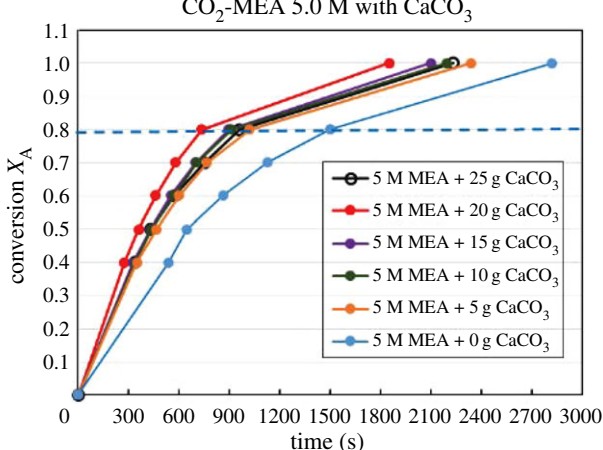

**Figure 6.** Catalytic $CO_2$ absorption of 5.0 M MEA solvents with $CaCO_3$ conversion ($X_A$) versus contact time ($t$).

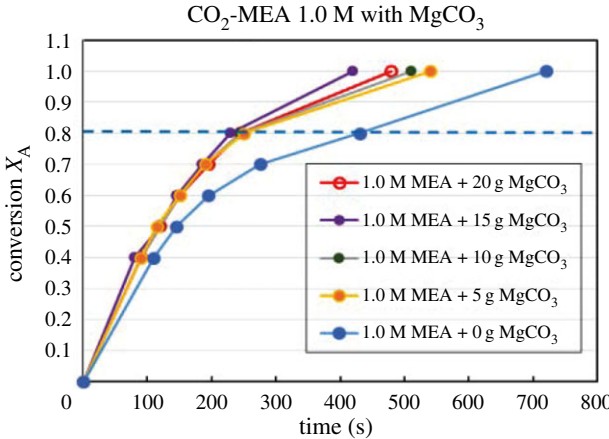

**Figure 7.** Catalytic $CO_2$ absorption of 1.0 M MEA solvents with $MgCO_3$ conversion ($X_A$) versus contact time ($t$).

Based on equation (3.5), $r_1 = K_B[A]$. This equation also indicated that the heterogeneous catalytic $CO_2$ absorption was the pseudo-first-order with respect to [MEA], the same as non-catalytic absorption. The rate of heterogeneous catalytic $CO_2$–MEA absorption was equation (4.1), similar to equation (2.4) of non-catalytic absorption. Despite the same reaction orders, the mechanisms and rate constants were different

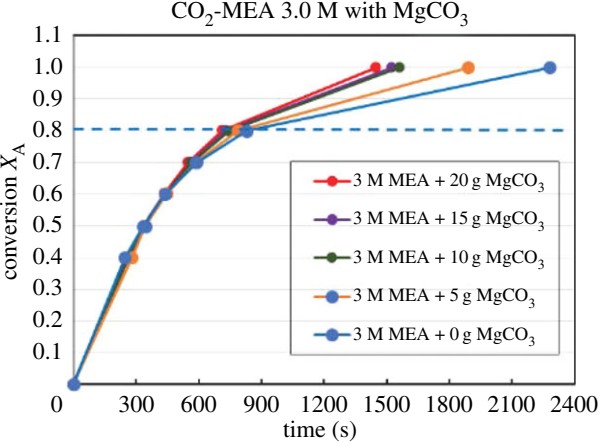

**Figure 8.** Catalytic $CO_2$ absorption of 3.0 M MEA solvents with $MgCO_3$ conversion ($X_A$) versus contact time ($t$).

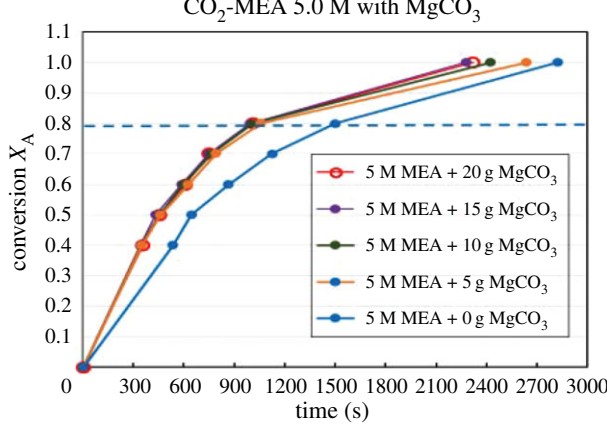

**Figure 9.** Catalytic $CO_2$ absorption of 5.0 M MEA solvents with $MgCO_3$ conversion ($X_A$) versus contact time ($t$).

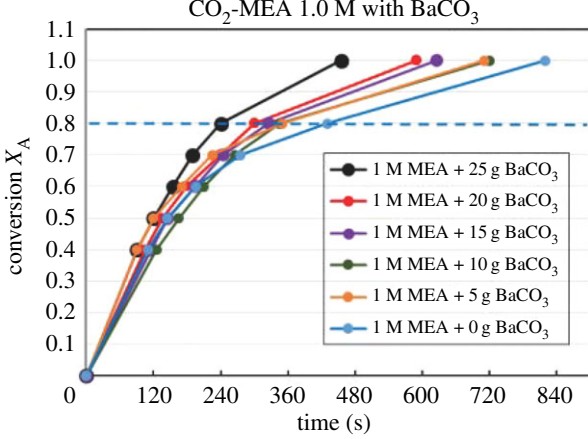

**Figure 10.** Catalytic $CO_2$ absorption of 1.0 M MEA solvent with $BaCO_3$ conversion ($X_A$) versus contact time ($t$).

under catalytic and non-catalytic absorptions. The reaction order of [MEA] can be directly extracted from graphical method from experimental data [2], but the power law model is a straightforward but over-simplified method lacking detailed intrinsic reaction mechanism and elementary steps [2].

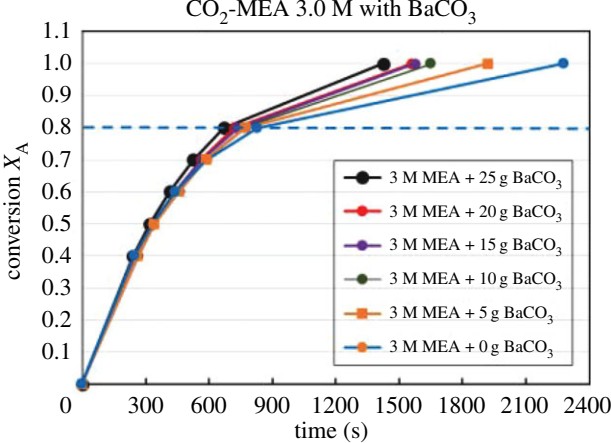

**Figure 11.** Catalytic $CO_2$ absorption of 3.0 M MEA solvent with $BaCO_3$ conversion ($X_A$) versus contact time ($t$).

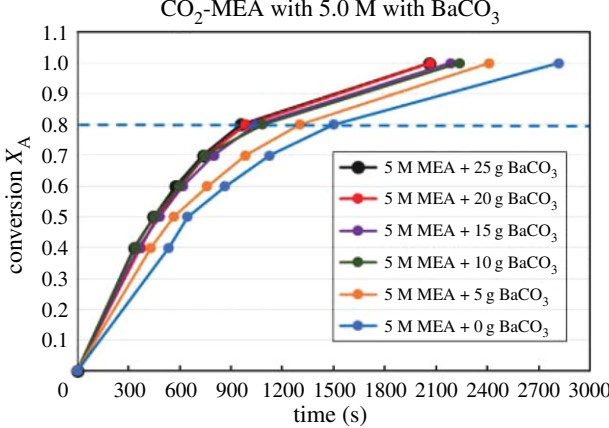

**Figure 12.** Catalytic $CO_2$ absorption of 5.0 M MEA solvent with $BaCO_3$ conversion ($X_A$) versus contact time ($t$).

The derivation of apparent rate law [26] was hard and time-consuming, but it accurately verified the mechanism and elementary steps.

$$r_{CO_2} = k_B[R_1R_2NH][CO_2] \tag{4.1}$$

$$r_{CO_2} = k_{2,R_1R_2NH}^Z[R_1R_2NH][CO_2]. \tag{2.4} \quad [2]$$

If the definition of rate equation (3.4) was combined with equations (2.4) and (4.1), the overall rate equation was equation (4.2), along with the differential integrated rate equation listed as equations (4.3) and (4.4). The $k[CO_2]$ were the slopes of figures 6–8.

$$r_A = -\frac{dC_A}{dt} = -\frac{dC_{A0}(1 - X_A)}{dt} = \frac{C_{A0}dX_A}{dt} = k[CO_2]C_A \tag{4.2}$$

$$-\frac{dC_A}{C_A} = k[CO_2]dt \tag{4.3} \text{ (Differentiation)}$$

$$\ln\left(\frac{1}{1 - X_A}\right) = k[CO_2]\,t. \tag{4.4} \text{ (Integration)}$$

Therefore, the slope of figures 6–8 ($X_A < 0.80$) was $k_B[CO_2]$. If we applied equation (3.3) to data ($X_A$, $t$) of non-catalytic $CO_2$ absorption (electronic supplementary material, table SA.1–SA.3), and plotted the similar curve as figures 6–8, the slope would be $kz[CO_2]$. Therefore, the ratio of slopes represented $k_B/kz$ based on equations (3.6) and (2.1), reflecting the enhancement of catalytic absorption over non-catalytic absorption. The ratios are categorized in electronic supplementary material, table SB.1.

**Table 3.** The rate model verification of RDS = B1 and B2 under each sub-case.

| Eley−Rideal models | | | solid catalysts | | |
|---|---|---|---|---|---|
| RDS | | rate equations | CaCO$_3$ | MgCO$_3$ | BaCO$_3$ |
| **B1** | $r_1$ | $r_1 = K_B \dfrac{[A]}{1 + k_4[C]} \approx K_B \dfrac{[A]}{k_4[C]}$ | | | |
| amine adsorption controlled | | no. of lines | **NO** | **NO** | **NO** |
| | | $R^2 > 0.990$ | 0/12 | 0/12 | 0/15 |
| | | $R^2 < 0.990$ | 12 | 12 | 15 |
| | integration | $\ln \dfrac{1}{1 - X_A} - X_A = K' t$ | | | |
| | $r_1$ | $r_1 = K_B \dfrac{[A]}{1 + k_4[C]} \approx K_B[A]$ | | | |
| | | no. of lines | **YES** | **YES** | **YES** |
| | integration | $\ln \dfrac{1}{1 - X_A} = K_B t$ | | | |
| | | $R^2 > 0.990$ | 10/12 | 12/12 | 14/15 |
| | | $R^2 < 0.990$ | 2 | 0 | 1 |
| **B2** | rate | $r_2 = K_B \left\{ \dfrac{[A]p_B}{K_1 + K_1 K_4[C] + [A]} \right\}$ | CaCO$_3$ | MgCO$_3$ | BaCO$_3$ |
| | | **ka = 0.05** | | | |
| Zwitterion formation controlled | | $\approx K_B \left\{ \dfrac{[A]p_B}{K_1 K_4[C] + [A]} \right\}$ | | | |
| | | no. of lines | **NO** | **NO** | **NO** |
| | | $R^2 > 0.990$ | 0/12 | 0/12 | 0/15 |
| | | $R^2 < 0.990$ | 12 | 12 | 15 |
| | | Ka = 0.5$k_1 k_4$ | $R^2$ | 0.95−0.98 | 0.95−0.97 | 0.95−0.97 |

(Continued.)

**Table 3.** (*Continued.*)

| Eley−Rideal models | | solid catalysts | | | |
|---|---|---|---|---|---|
| RDS | rate equations | CaCO$_3$ | MgCO$_3$ | BaCO$_3$ |
| | $k_a \left( \ln \dfrac{1}{1 - X_A} - X_A \right) + X_A = K'' t$ | ***ka* = 0.005** | | | |
| | | no. of lines | **NO** | **NO** | **NO** |
| | | $R^2 > 0.990$ | 0/12 | 0/12 | 0/15 |
| | | $R^2 < 0.990$ | 12 | 12 | 15 |
| | | $R^2$ | 0.93−0.96 | 0.94−0.96 | 0.91−0.96 |
| | $X_A = K'' t$ | ***ka* = 0** | | | |
| | | no. of lines | **NO** | **NO** | **NO** |
| | | $R^2 > 0.990$ | 0/12 | 0/12 | 0/15 |
| | | $R^2 < 0.990$ | 12 | 12 | 15 |
| | | $R^2$ | 0.93−0.97 | 0.94−0.97 | 0.94−0.97 |

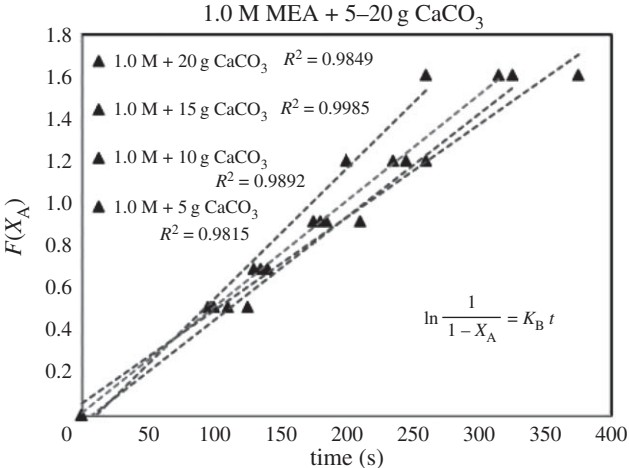

**Figure 13.** $F(X_A)$ versus $t$ for 1.0 M MEA with $CaCO_3$.

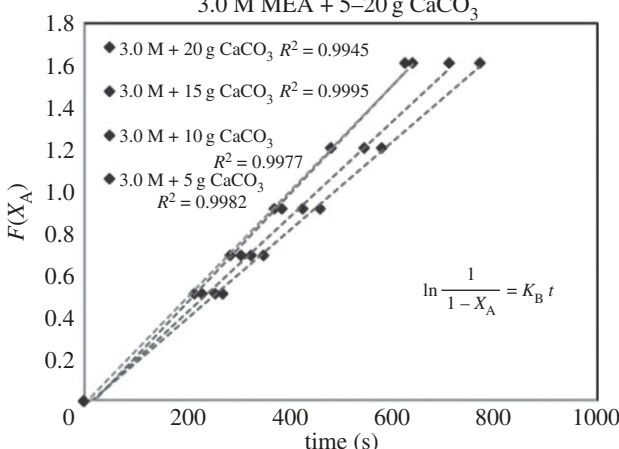

**Figure 14.** $F(X_A)$ versus $t$ for 3.0 M MEA with $CaCO_3$.

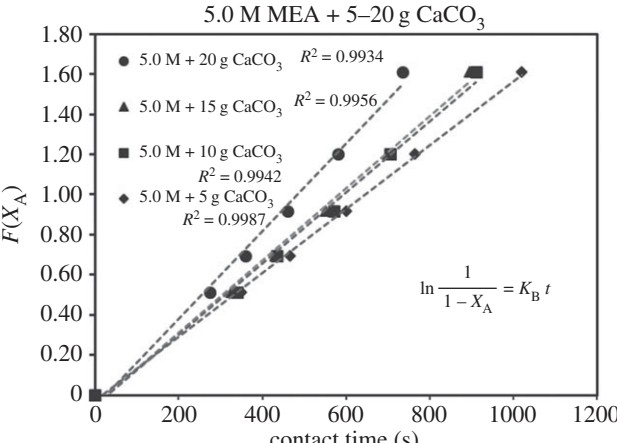

**Figure 15.** $F(X_A)$ versus $t$ for 5.0 M MEA with $CaCO_3$.

Based on electronic supplementary material, table SB.1, slopes of catalytic absorption were steeper than that of the non-catalytic one. The enhanced catalysts were about 20–100% higher for $CaCO_3$, 20–80% higher for $MgCO_3$, and 25–80% higher for $BaCO_3$. The optimized catalysis was 100% with

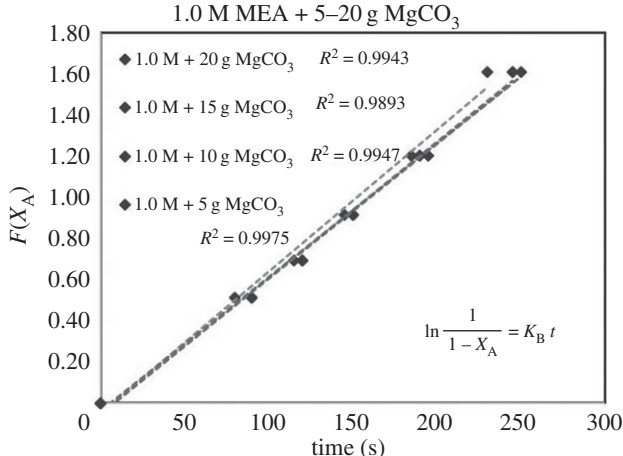

**Figure 16.** $F(X_A)$ versus $t$ for 1.0 M MEA with $MgCO_3$.

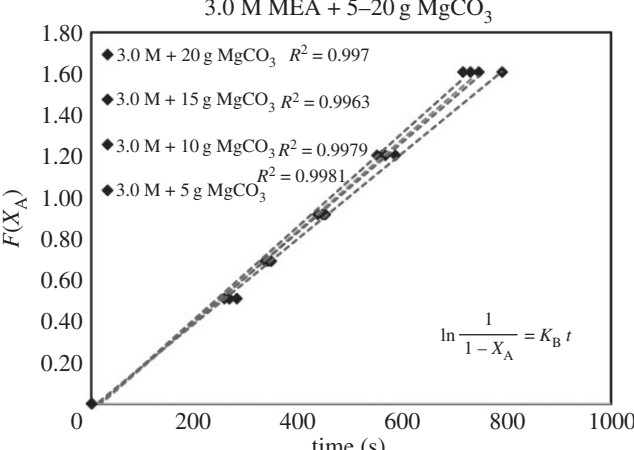

**Figure 17.** $F(X_A)$ versus $t$ for 3.0 M MEA with $MgCO_3$.

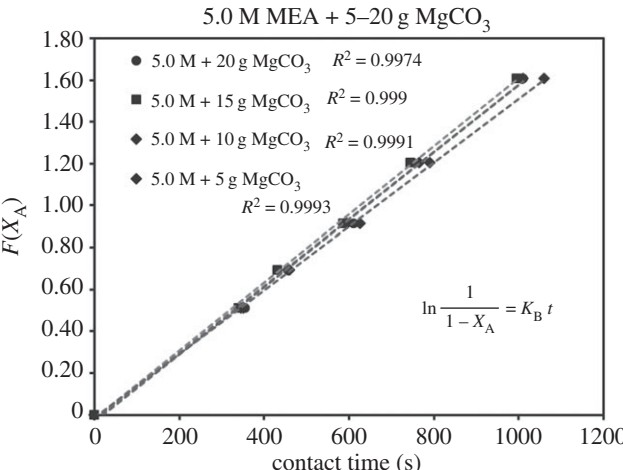

**Figure 18.** $F(X_A)$ versus $t$ for 5.0 M MEA with $MgCO_3$.

20 g $CaCO_3$ and 5 M MEA. Such improvements resulted from increased surface areas and active sites that enhanced amine adsorption and reduced activation energy Ea.

However, the experimental process was rather limited [23], and it was only adequate to verify the rate model as a start-up. This Eley–Rideal model still awaits much further analyses with updated experimental apparatus

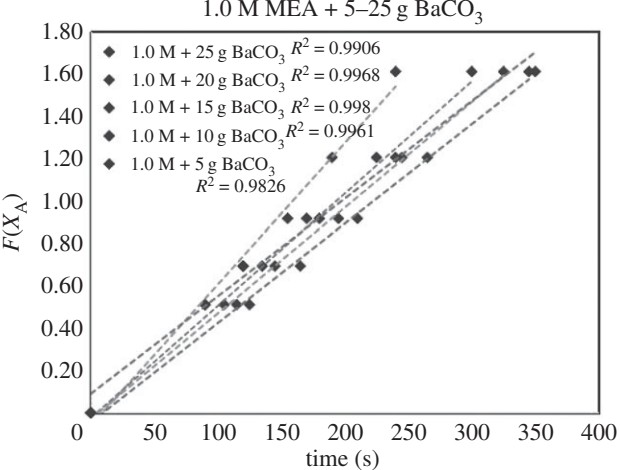

**Figure 19.** $F(X_A)$ versus $t$ for 1.0 M MEA with BaCO$_3$.

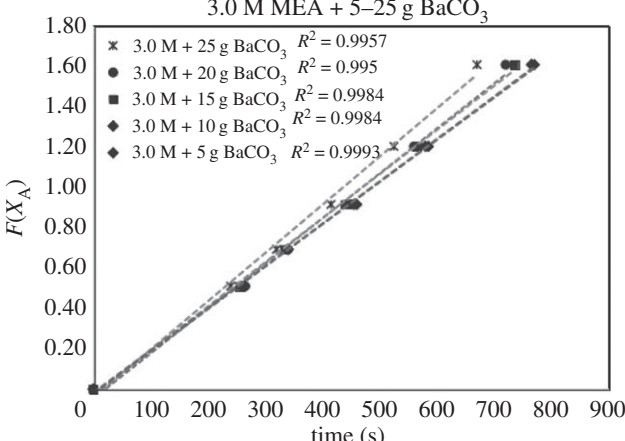

**Figure 20.** $F(X_A)$ versus $t$ for 3.0 M MEA with BaCO$_3$.

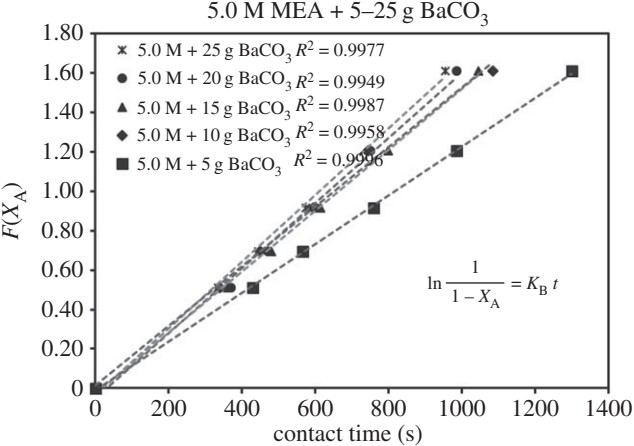

**Figure 21.** $F(X_A)$ versus $t$ for 5.0 M MEA with BaCO$_3$.

such as GC-MS HPLC for the analysis of reaction products, molecular simulations and comprehensive mathematic simulations [2]. The molecular simulation needs to be calculated of catalytic CO$_2$–MEA reactions with the existence of CaCO$_3$ with density function theory (DFT) to discover Ea of catalytic reactions. The kinetic analysis of another solid base 'KMgO/CNT (carbon nano-tubes)' with ER model will be future work, since this material was reported to be effective for CO$_2$–MEA absorption recently [31].

# 5. Conclusion

The Eley–Rideal model was proposed for catalytic carbamate formation of $CO_2 + RR'NH$ with $MCO_3$, based on the similar reactions of $CO_2 + MeOH$ over $ZrO_2$–$MgO$ catalyst [25]. For the case of MEA($RNH_2$), the rate-determining step was B1 of amine adsorption. It was the pseudo-first-order for [MEA]. The heterogeneous catalysis enlarged the second rate constant ($kz$) of MEA to 20–100% higher. For DEA, the rate-determining step was among B1–B4, but it required intensive literature studies and experimental results based on highly accurate kinetic results to verify the equations properly.

Solid surface contained much bigger gas–liquid interfacial area with abundant active sites, resulting in the enhancement and facilitation of molecule mass transfer and the reduction of activation energy Ea. The reduction of activation energy Ea was more important in accelerating reaction rates. The effect of mass transfer could be calculated from $k_{Gav}$ on the basis of experiments, and Ea could be calculated from either molecular simulation or Arrhenius equation based on experimental data of kinetics analyses.

Ethics. We received ethical approval from a 'lab safety and research ethics committee' of University of Shanghai for Science and Technology to carry out our study. This university provides consent in a research ethics and code of lab safety and ethics in scientific research and publication.

Data accessibility. The datasets supporting this article have been uploaded as part of the electronic supplementary material, A and B; including mechanism development (A) and experimental data (B).

Authors' contributions. H.S. contributed to conception and design, analysis and interpretation of data; the Eley–Rideal model was introduced into the field, and the manuscript revision were completed. H.S. also drafted the manuscript, together with other corresponding authors Y.H. and L.J. M.H. contributed to the experimental data of $CO_2$ absorptions figures 3–5. Y.H. and L.C. provided background concept of figure 1, elementary steps on solid surface; and they helped the development of the steps in support information A in the electronic supplementary material. L.Z. helped to completed the linear regression of data figures 3–5 (support information B in the electronic supplementary material), and generated figures 6–8 with help. L.J. provided the results of figure 2, SEM and BET with instruments. M.C. provided the high-resolution figures 1–10 for the manuscript. H.I. and P.T. completed the final approval of the version to submit after several intensive revisions. H.S. agreed to be accountable for all aspects of the work in ensuring that questions related to the accuracy or integrity of any part of the work.

Competing interests. We declare we have no competing interests.

Funding. This work was supported by the National Natural Science Foundation of China (NSFC nos. 21606150, 61775139), Young Eastern Scholar (QD 2016011), with complement (QD 2016011–005–11). Natural Sciences and Engineering Research Council of Canada (NSERC) are acknowledged.

Acknowledgements. We also thank Yunlong Zhou for completion of original experiments with original data but did not meet the authorship criteria.

# Appendices

The integrated analysis of equations (3.3)–(3.5), the rate model was developed based on steady-state approximation.

CASE I: RDS = B1, where $r_2 = r_3 = r_4 = 0$

$$r_1 = K_B\left\{\frac{[A]}{1 + K_4[C]}\right\}; \tag{B1}$$

$Cmate_{(s)} = K_4[C]C_{(s)}$ (B4), $r_4 = 0$ (steady-state approximation)
Sub-case I: $K_4[C] = \theta_{carbamate}/\theta_{empty} \ll 1$; abundant empty active sites

$$r_1 = K_B [A] \quad \text{Pseudo-first-order of [MEA]} \tag{B5}$$

$$r_1 = C_{A0}\frac{dX_A}{dt} = K_B C_{A0}(1 - X_A) \tag{B6}$$

$$\frac{dX_A}{dt} = K_B(1 - X_A) \quad \therefore \rightarrow \frac{dX_A}{1 - X_A} = K_B\,dt \tag{B7}$$

$$\int_0^X \frac{1}{1 - X_A}dX_A = \int_0^t K_B\,dt \tag{B8}$$

and

$$\ln\frac{1}{1 - X_A} = K_B\,t. \tag{B9}$$

Sub-case II: $K_4[C] = \theta_{\text{carbamate}}/\theta_{\text{empty}} \gg 1$; few empty active sites

$$r_1 = K_B\left\{\frac{[A]}{K_4[C]}\right\}$$

$$r_1 = C_{A0}\frac{dX_A}{dt} = \frac{2K_B}{K_4}\frac{1-X_A}{X_A} \tag{B10}$$

$$\frac{dX_A}{dt} = \frac{2K_B}{K_4 C_{A0}}\frac{1-X_A}{X_A} \quad \therefore \rightarrow \quad \frac{X_A\,dX_A}{1-X_A} = \frac{2K_B}{K_4 C_{A0}}\,dt = K'dt \tag{B11}$$

$$\int_0^X \frac{X_A}{1-X_A}dX_A = \int_0^t K'\,dt \tag{B12}$$

and

$$\ln\frac{1}{1-X_A} - X_A = K'\,t \qquad \frac{2K_B}{K_4 C_{A0}} = K'. \tag{B13}$$

CASE II, RDS = B2 ($r_2$); where $r_1 = r_3 = r_4 = 0$

$$r_2 = K_B\left\{\frac{[A]p_B}{K_1 + K_1 K_4[C] + [A]}\right\}; \text{ [B2] where } p_B = pCO_2 = 1 \text{ (Pure } CO_2).$$

From steady state of B1, $r_1 = 0$:

$$K_1 = \frac{[A]}{Am_{(s)}}C_{(s)} \tag{B1}$$

$$r_2 = K_B\left\{\frac{[A]p_B}{K_1 K_4[C] + [A]}\right\} \quad \text{[B2], replace } [A],[C] \text{ with } X_A$$

$$r_2 = K_B p_B \frac{[A]}{K_1 K_4[C] + [A]} = \frac{K_B p_B}{1}\frac{1-X_A}{(1/2)K_1 K_4 X_A + [1-X_A]} \tag{B14}$$

$$r_2 = C_{A0}\frac{dX_A}{dt} = \frac{K_B p_B}{1}\frac{1-X_A}{(1/2)K_1 K_4 X_A + [1-X_A]} \tag{B15}$$

and

$$\frac{\{(1/2)k_1 k_4 X_A + [1-X_A]\}\,dX_A}{1-X_A} = \frac{K_B p_B}{C_{A0}}\,dt. \tag{B16}$$

Set: $K'' = (K_B p_B)/(C_{A0})$ and $K_a = (1/2)K_1 K_4$ (B16) into (B15)

$$\frac{\{k_a X_A + [1-X_A]\}\,dX_A}{1-X_A} = k_a\frac{X_A}{1-X_A}dX_A + 1\,dX_A = K''\,dt \tag{B17}$$

$$k_a\int_0^X \frac{X_A}{1-X_A}\,dX_A + \int_0^X dX_A = \int_0^t K'. \tag{B18}$$

After integration with conversion rate $X_A$, the equation was listed for RDS = B2:

$$k_a\left\{\ln\frac{1}{1-X_A} - X_A\right\} + X_A = K''\,t. \tag{B19}$$

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
