## [Reviewer comments · Royal Society Open Science]

Review History

RSOS-181917.R0 (Original submission)

Review form: Reviewer 1

Is the manuscript scientifically sound in its present form?

No

Are the interpretations and conclusions justified by the results?

No

Is the language acceptable?

No

Is it clear how to access all supporting data?

No

Do you have any ethical concerns with this paper?

No

Have you any concerns about statistical analyses in this paper?

Yes

Recommendation?

Reject

Comments to the Author(s)

The manuscript describes a study of the reaction mechanisms for the reaction of adsorbed ethanol amine and carbon dioxide with the assistance of primary amines and secondary amines. The authors explored different rate-determining steps in their proposed mechanism to elucidate different potential overall rate laws. They compared the derived rate laws with experimental data to determine possible rate-determining step in the proposed mechanism. It is refreshing to read a fundamental study on understanding the chemical reactions. Overall, the manuscript needs major re-writing. It is very difficult to follow the ideas in the manuscript because a lot of the math symbols were either ill-defined or changed in the middle of the derivations. There are errors in some of the math derivations. The criteria in the fittings of the experimental data is unexplained. In summary, the reviewer does not recommend this manuscript for publication, but would encourage the authors to resubmit the manuscript after major revision. Below are more detailed comments:

1. It was unclear to the reviewer that ethanol amine was preadsorbed to the metal carbonates in the manuscript writing. It was revealed by the graphics displaying the reaction schematic.
2. The authors proposed Eley-Rideal model to explain the reaction mechanism. While the proposed idea is possible, the authors incorrectly stated that reference 25 claimed that their reactions followed Eley-Rideal reactions. The reviewer read reference 25 and found that it claimed Langmuir-Hinshelwood mechanism instead. The authors copied part of the figure in reference 25 and discussed it in the supporting information. They should cite the materials clearly.
3. The manuscript mentioned a lot of unusual terms. The reviewer does not understand the meaning of these terms. For example, what is liquid [OH-]?
4. Many of the mathematics symbols were not defined in the main text. It is difficult to follow the discussions. For example, on page 5, line 24, what are k (superscript: dot) (subscript: OH-) and r (superscript: dot) (subscript: OH-)?
5. Why was BaCO₃ chosen for the evaluation?
6. Page 7, line 45, what is figure 1a?
7. It is unclear to the reviewer why r_3 is zero for B₃ when B₁ is the rate determining step. According to the elementary step, B₃ is a one direction elementary step reaction.
8. There are many typos in the manuscript. For example, "concertation" on page 9, line 16.
9. There is an error for equation (12). It should be $[C] = CA_0 \times \alpha = \frac{1}{2} CA_0 (1 - X_A)$. The manuscript lacks clear derivations of the rate equations.
10. The manuscript lacks detailed experimental procedures for reproducing the materials.
11. The authors should justify why they chose $R^2 > 0.990$ as the criteria for good fittings of the data.
12. Why did the authors present no characterization data for BaCO₃?
13. What are the error bars for the experimental data?
14. It is difficult to evaluate the rate law by fitting the $F(X_A)$ vs time. It is better to directly extract the order of reactions with respect to each reactants from the experimental data.

Review form: Reviewer 2

Is the manuscript scientifically sound in its present form?

Yes

Are the interpretations and conclusions justified by the results?

Yes

Is the language acceptable?

Yes

Is it clear how to access all supporting data?

Not Applicable

Do you have any ethical concerns with this paper?

No

Have you any concerns about statistical analyses in this paper?

No

Recommendation?

Accept with minor revision (please list in comments)

Comments to the Author(s)

This manuscript proposed a new model for heterogeneous catalytic CO₂-RR'NH interactions. The reaction process, the elementary steps, and rate limiting steps was developed. These results are useful for further studies with secondary amines etc. This manuscript is worth publication.

Several small problems need to be edited to improve the quality:

1. Introduction:

(1) It is mentioned mostly that the reaction of CO₂ with primary amine and secondary amine, but little of CO₂ with tertiary amine. Please mention the reactions with tertiary amine to complete the full map of kinetics.

2. Theory

(2) In Section 2.2 of reaction (B2), the author mentioned the activation energy (E_a) of carbamate formation. The E_a of non-catalytic reaction can be calculated from rate constant from kinetic analysis. How to obtain the E_a of catalytic reaction ?

(3) Why these elementary steps B1-B4 unsuitable for tertiary amine ? Please explain briefly.

3. Result and Discussion

(4) First paragraph, "The solid surface had abundant empty active sites with very few carbamate adsorbed." Please provide a detailed explanation of this sentence.

(5) Last paragraph, "This Eley-Rideal model still awaits much further analyses with updated experimental apparatus", what types of experiments did you wish to conduct as the next task?

4. Conclusions

(6) For catalytic CO₂-DEA reaction with this model, is RDS still B1 ?

(7) Last sentence, the increased reaction rates is result in either mass transfer or reduced activation energy. Which part is dominant and How to distinguish these effects ?

Decision letter (RSOS-181917.R0)

21-Jan-2019

Dear Dr Shi:

Manuscript ID: RSOS-181917

Title: "Eley-Rideal model of heterogeneous catalytic carbamate formation based on CO₂-MEA absorptions with CaCO₃, MgCO₃ and BaCO₃"

Thank you for submitting the above manuscript to Royal Society Open Science. Your paper was sent to reviewers and their comments are included at the bottom of this letter.

In view of the concerns raised by the reviewers, the manuscript has been rejected in its current form. However, a new manuscript may be submitted which takes into consideration these comments.

Please note that resubmitting your manuscript does not guarantee eventual acceptance, and that your resubmission will be subject to peer review before a decision is made.

Your resubmitted manuscript should be submitted by 21-Jul-2019. If you are unable to submit by this date please contact the Editorial Office.

On behalf of the Subject Editor Professor Anthony Stace and the Associate Editor Professor Hazel Cox

REVIEWER(S) REPORTS:
Associate Editor Comments to Author ():
RSC Associate Editor:
Comments to the Author:
(There are no comments.)

RSC Subject Editor:
 Comments to the Author:
 (There are no comments.)

Reviewers' Comments to Author:
 Reviewer: 1

Comments to the Author(s)

The manuscript describes a study of the reaction mechanisms for the reaction of adsorbed ethanol amine and carbon dioxide with the assistance of primary amines and secondary amines. The authors explored different rate-determining steps in their proposed mechanism to elucidate different potential overall rate laws. They compared the derived rate laws with experimental data to determine possible rate-determining step in the proposed mechanism. It is refreshing to read a fundamental study on understanding the chemical reactions. Overall, the manuscript needs major re-writing. It is very difficult to follow the ideas in the manuscript because a lot of the math symbols were either ill-defined or changed in the middle of the derivations. There are errors in some of the math derivations. The criteria in the fittings of the experimental data is unexplained. In summary, the reviewer does not recommend this manuscript for publication, but would encourage the authors to resubmit the manuscript after major revision. Below are more detailed comments:

1. It was unclear to the reviewer that ethanol amine was preadsorbed to the metal carbonates in the manuscript writing. It was revealed by the graphics displaying the reaction schematic.
2. The authors proposed Eley-Rideal model to explain the reaction mechanism. While the proposed idea is possible, the authors incorrectly stated that reference 25 claimed that their reactions followed Eley-Rideal reactions. The reviewer read reference 25 and found that it claimed Langmuir-Hinshelwood mechanism instead. The authors copied part of the figure in reference 25 and discussed it in the supporting information. They should cite the materials clearly.
3. The manuscript mentioned a lot of unusual terms. The reviewer does not understand the meaning of these terms. For example, what is liquid [OH-]?
4. Many of the mathematics symbols were not defined in the main text. It is difficult to follow the discussions. For example, on page 5, line 24, what are k (superscript: dot) (subscript: OH-) and r (superscript: dot) (subscript: OH-)?
5. Why was BaCO₃ chosen for the evaluation?
6. Page 7, line 45, what is figure 1a?
7. It is unclear to the reviewer why r_3 is zero for B₃ when B₁ is the rate determining step. According to the elementary step, B₃ is a one direction elementary step reaction.
8. There are many typos in the manuscript. For example, "concertation" on page 9, line 16.
9. There is an error for equation (12). It should be $[C] = CA_0 \times a = \frac{1}{2} CA_0 (1 - X_A)$. The manuscript lacks clear derivations of the rate equations.
10. The manuscript lacks detailed experimental procedures for reproducing the materials.
11. The authors should justify why they chose $R^2 > 0.990$ as the criteria for good fittings of the data.
12. Why did the authors present no characterization data for BaCO₃?
13. What are the error bars for the experimental data?
14. It is difficult to evaluate the rate law by fitting the $F(X_A)$ vs time. It is better to directly extract the order of reactions with respect to each reactants from the experimental data.

Reviewer: 2

Comments to the Author(s)

This manuscript proposed a new model for heterogeneous catalytic CO₂-RR'NH interactions. The reaction process, the elementary steps, and rate limiting steps was developed. These results are useful for further studies with secondary amines etc. This manuscript is worth publication.

Several small problems need to be edited to improve the quality:

1. Introduction:

(1) It is mentioned mostly that the reaction of CO₂ with primary amine and secondary amine, but little of CO₂ with tertiary amine. Please mention the reactions with tertiary amine to complete the full map of kinetics.

2. Theory

(2) In Section 2.2 of reaction (B2), the author mentioned the activation energy (E_a) of carbamate formation. The E_a of non-catalytic reaction can be calculated from rate constant from kinetic analysis. How to obtain the E_a of catalytic reaction ?

(3) Why these elementary steps B1-B4 unsuitable for tertiary amine ? Please explain briefly.

3. Result and Discussion

(4) First paragraph, "The solid surface had abundant empty active sites with very few carbamate adsorbed." Please provide a detailed explanation of this sentence.

(5) Last paragraph, "This Eley-Rideal model still awaits much further analyses with updated experimental apparatus", what types of experiments did you wish to conduct as the next task?

4. Conclusions

(6) For catalytic CO₂-DEA reaction with this model, is RDS still B1 ?

(7) Last sentence, the increased reaction rates is result in either mass transfer or reduced activation energy. Which part is dominant and How to distinguish these effects ?

Author's Response to Decision Letter for (RSOS-181917.R0)

See Appendix A.

RSOS-190311.R0

Review form: Reviewer 1

Is the manuscript scientifically sound in its present form?

Yes

Are the interpretations and conclusions justified by the results?

Yes

Is the language acceptable?

Yes

Is it clear how to access all supporting data?

Not Applicable

Do you have any ethical concerns with this paper?

No

Have you any concerns about statistical analyses in this paper?

I do not feel qualified to assess the statistics

Recommendation?

Accept with minor revision (please list in comments)

Comments to the Author(s)

The authors answer most of the questions from reviewers. There are two minor but necessary modifications needed:

1. In the described mechanisms, the arrows showed the direction of the molecule (such as CO₂) attacking the adsorbed amines. This counters the common convention that the arrows indicate the flow direction of the electrons. Using this example, the lone pair electrons from the N of the amine should attack the carbon of the CO₂ to form the C-N bond. So under the usual convention, the direction of the arrow should go from the amine to the carbon of the CO₂. The authors should use the typical convention instead.
2. There are many typos, random capitalization of words and missed subscripts in the manuscript and the references. There are too many of them to be listed here. These to be corrected.

Review form: Reviewer 2

Is the manuscript scientifically sound in its present form?

Yes

Are the interpretations and conclusions justified by the results?

Yes

Is the language acceptable?

Yes

Is it clear how to access all supporting data?

Yes

Do you have any ethical concerns with this paper?

No

Have you any concerns about statistical analyses in this paper?

No

Recommendation?

Accept as is

Comments to the Author(s)

The author did responded to all the questions, and provided reasonable explanations. The changes were appropriate and helpful for the readers to understand the manuscript.

Decision letter (RSOS-190311.R0)

19-Mar-2019

Dear Dr Shi:

Title: Eley-Rideal model of heterogeneous catalytic carbamate formation based on CO₂-MEA absorptions with CaCO₃, MgCO₃ and BaCO₃
Manuscript ID: RSOS-190311

Thank you for submitting the above manuscript to Royal Society Open Science. On behalf of the Editors and the Royal Society of Chemistry, I am pleased to inform you that your manuscript will be accepted for publication in Royal Society Open Science subject to minor revision in accordance with the referee suggestions. Please find the reviewers' comments at the end of this email.

The reviewers and handling editors have recommended publication, but also suggest some minor revisions to your manuscript. Therefore, I invite you to respond to the comments and revise your manuscript.

Please also include the following statements alongside the other end statements. As we cannot publish your manuscript without these end statements included, if you feel that a given heading is not relevant to your paper, please nevertheless include the heading and explicitly state that it is not relevant to your work. We have included a screenshot example of the end statements for reference.

- Ethics statement

Please clarify whether you received ethical approval from a local ethics committee to carry out your study. If so please include details of this, including the name of the committee that gave consent in a Research Ethics section after your main text. Please also clarify whether you received informed consent for the participants to participate in the study and state this in your Research Ethics section.

OR

Please clarify whether you obtained the necessary licences and approvals from your institutional animal ethics committee before conducting your research. Please provide details of these licences and approvals in an Animal Ethics section after your main text.

OR

Please clarify whether you obtained the appropriate permissions and licences to conduct the fieldwork detailed in your study. Please provide details of these in your methods section.

- Acknowledgements

Because the schedule for publication is very tight, it is a condition of publication that you submit the revised version of your manuscript before 28-Mar-2019. Please note that the revision deadline will expire at 00.00am on this date. If you do not think you will be able to meet this date please let me know immediately.

Best wishes,

Dr Laura Smith
Publishing Editor, Journals

On behalf of the Subject Editor Professor Anthony Stace and the Associate Editor Professor Hazel Cox.

RSC Associate Editor
Comments to the Author:
(There are no comments.)

Reviewer comments to Author:
Reviewer: 2

Comments to the Author(s)

The author did responded to all the questions, and provided reasonable explanations. The changes were appropriate and helpful for the readers to understand the manuscript.

Reviewer: 1

Comments to the Author(s)

The authors answer most of the questions from reviewers. There are two minor but necessary modifications needed:

1. In the described mechanisms, the arrows showed the direction of the molecule (such as CO₂) attacking the adsorbed amines. This counters the common convention that the arrows indicate the flow direction of the electrons. Using this example, the lone pair electrons from the N of the amine should attack the carbon of the CO₂ to form the C-N bond. So under the usual convention, the direction of the arrow should go from the amine to the carbon of the CO₂. The authors should use the typical convention instead.
2. There are many typos, random capitalization of words and missed subscripts in the manuscript and the references. There are too many of them to be listed here. These to be corrected.

Author's Response to Decision Letter for (RSOS-190311.R0)

See Appendix B.

Decision letter (RSOS-190311.R1)

05-Apr-2019

Dear Dr Shi:

Title: Eley-Rideal model of heterogeneous catalytic carbamate formation based on CO₂-MEA absorptions with CaCO₃, MgCO₃ and BaCO₃
Manuscript ID: RSOS-190311.R1

It is a pleasure to accept your manuscript in its current form for publication in Royal Society

Open Science. The chemistry content of Royal Society Open Science is published in collaboration with the Royal Society of Chemistry.

RSC Associate Editor
Comments to the Author:
(There are no comments.)

Reviewer(s)' Comments to Author:

Appendix A

Reviewers' Comments with response

Reviewer 1:

Comments to the Author(s)

The manuscript describes a study of the reaction mechanisms for the reaction of adsorbed ethanol amine and carbon dioxide with the assistance of primary amines and secondary amines. The authors explored different rate-determining steps in their proposed mechanism to elucidate different potential overall rate laws. They compared the derived rate laws with experimental data to determine possible rate-determining step in the proposed mechanism. It is refreshing to read a fundamental study on understanding the chemical reactions.

Overall, the manuscript needs major re-writing. It is very difficult to follow the ideas in the manuscript because **a lot of the math symbols were either ill-defined or changed in the middle of the derivations. There are errors in some of the math derivations. The criteria in the fittings of the experimental data is unexplained.** In summary, the reviewer does not recommend this manuscript for publication, but would encourage the authors to **resubmit the manuscript after major revision.**

Response:

The reviewer has raised many useful points, and we are very grateful for his encouragements. The kinetics of $\text{CO}_2\text{-R}_1\text{R}_2\text{NH}$ in aqueous solution has been intensively studied for decades with massive publications since 1968. [2,4-21] A recent review [2] (2012) comprehensively described these kinetics models with over 100 references. However, the heterogeneous catalysts of $\text{CO}_2\text{-R}_1\text{R}_2\text{NH}$ has just been discovered since 2013. Several publications have reported experimental results and verified the catalysis, but there are very little precedent studies of heterogeneous in terms of “kinetic models with reaction mechanism”. It is very difficult to develop these kinetic models with “mechanistic study”, including reaction mechanism, elementary steps, rate determining steps and rate equations. Therefore, we have no choice but to develop the Apparent Rate Law from Steady State approximation [26] and verify reaction kinetics the model with experimental data: the definition of reaction rates “ $r = dCA/dt$ ” with Integrated Analysis [27].

The investigations are time-consuming and require efforts on the kinetic model development and mathematic equations. We have studied a large number of kinetic studies of non-catalytic $\text{CO}_2\text{-MEA}$ (DEA) reactions since 1968. [4-21] We managed to generate massive rate equations within months. It is difficult to generate rate equations and verify them with experimental data.

As mentioned, the math equations were explained with reference, and some symbols were checked and modified in Support Information. The details of experimental data fitting were explained.

We will try our best to address these problems and fix the manuscript properly.

Below are more detailed comments:

Question 1. Operational conditions of Experiment

1. It was unclear to the reviewer that ethanol amine was pre-adsorbed to the metal carbonates in the manuscript writing. It was revealed by the graphics displaying the reaction schematic.

Response:

The reviewer's suggestion was well taken. The experimental procedures were indeed oversimplified. This process was similar to that of other study [23], which focused on CO₂-DEA absorption with the aid of CaCO₃ and MgCO₃. The details were neglected to avoid duplication.

Fig. S1. Stirred Cell Reactor for Catalytic CO₂-MEA interactions with a water scrubbing process.

A set of stirred-cell reactor was built with suspension of pelletized chemicals for the experiments, and the internal diameter of the reactor was 8.4 cm, with a constant interfacial area of 55.4 cm². The solids were wrapped into two balls and suspended onto the gas-liquid interface. The intersection area of two solid balls was about 9.8 cm² (2.5 cm in diameter for each). The amine solvents were pre-filled into the reactor, with part of the MEA molecules pre-adsorbed onto the solid surface. The reactor was placed in a cool water bath with a magnetic stirrer. The CO₂ flowrate was in the range of 1.0-1.5 L/min. The thermometer was placed inside the reactor to detect the reaction temperature. Pure CO₂ was introduced to a water-scrubbing process and then flowed into the batch reactor containing amine solvents with bubbles.

During the catalytic absorption process, the solid chemicals were wrapped and placed at the gas-liquid interface within the stirred cell reactor first. Then the gas CO₂ (100%) was introduced into the reactor for reaction. Hence, part of the MEA was pre-absorbed onto the solid surface before reacting with CO₂ for catalytic absorption.

After introduced into amine solvents, CO₂ reacted with MEA solvent via both non-catalytic and

catalytic reaction pathways. For the non-catalytic pathway, CO₂ directly reacted with MEA solvent, and the kinetics had been intensively studied. For the catalytic pathway, the CO₂ reacted with the MEA molecules pre-adsorbed onto the solid surface. The kinetics was the focus of this study.

Changes: Section 3.1 added paragraph 2, 3 (Same as Question 10)

The CO₂ absorption process was similar to other works. [23] A set of stirred-cell reactor was built with suspension of pelletized chemicals for the experiments, and the internal diameter of the reactor was 8.4 cm, with a constant interfacial area of 55.4 cm². The solids were wrapped into two balls and suspended onto the gas-liquid interface. The intersection area of two solid balls was about 9.8 cm² (2.5 cm in diameter for each). The amine solvents were pre-filled into the reactor, with part of the MEA solvents pre-adsorbed onto the solid surface. The reactor was placed in a cool water bath with a magnetic stirrer. The CO₂ flowrate was in the range of 1.0-1.5 L/min. A thermometer was placed inside the reactor to detect the temperature. Pure CO₂ was introduced to a water-scrubbing process and then flowed into the batch reactor containing amine solvents with bubbles.

After introduced into amine solvents, the CO₂ reacted with MEA solvent via both non-catalytic and catalytic reaction pathways. For the non-catalytic pathway, the CO₂ directly reacted with MEA solvent, and the kinetics had been intensively studied.[2] For the catalytic pathway, the CO₂ reacted with the MEA molecules pre-adsorbed onto the solid surface, which was the focus of this study.

Question 2. The Role of Reference 25 and the application

2. The authors proposed Eley-Rideal model to explain the reaction mechanism. While the proposed idea is possible, the authors incorrectly stated that reference 25 claimed that their reactions followed Eley-Rideal reactions. **The reviewer read reference 25 and found that it claimed Langmuir-Hinshelwood mechanism instead.** The authors copied part of the figure in reference 25 and discussed it in the supporting information. They should cite the materials clearly.

Response:

The reviewer raised a very important question, and we re-investigate the problems carefully.

We understand that reactions of CO₂ + MeOH to synthesis DMC (Ref. 25 in 2016) claimed Langmuir-Hinshelwood mechanism (LH). Within that publication, they also cited two earlier references (2003 and 2011), which claimed that the reaction was via Eley-Rideal mechanism (ER).

Whether this kinetics mechanism was LH or ER was determined by different types of solid catalysts, which was not an undisputed conclusion.

References of ER mechanism:

Eta V, Maki-Arvela P, Warna J, Salmi T, Mikkola JP, Murzin DY. 2011 *Applied Catalysis A*. **404**, 39-46. (doi:10.1016/j.apcata.2011.07.004).

S.A. Anderson, T.W. Root, *J. Catal.* 217 (2003) 396–405

References of LH mechanism:

H.J. Lee, W. Joe, J.C. Jung, I.K. Song, *Korean J. Chem. Eng.* 29 (2012) 1019–1024.

Martin CM, Li L, Bhalkikar A, Doyle JE, Zeng XC, Cheung CL. 2016 *Journal of Catalysis*. **340**, 295-301. (doi:10.1016/j.jcat.2016.06.003)

We studied the Results and Discussions in Ref [25] carefully. The authors could only “**suggest**” that the reactions “**may follow**” LH mechanism “**over ceria catalysts**”. The mechanism was verified by their experimental data, which was greatly affected by different catalysts.

The text of Ref [25]: “The discussed initial reaction rate kinetics experiments strongly suggest that the conversion of methanol and CO₂ to DMC **over ceria catalysts follows a Langmuir–Hinshelwood mechanism** where the CO₂ and methanol are binding to the catalyst in separate steps, consistent with previous kinetics studies modeling the reaction from the reaction profile [16]. Interestingly, this result contrasts with that shown for the production of DMC from methanol and carbon monoxide **over copper zeolite catalysts**, which were found to have a non-negative rate order with respect to methanol and **are believed to follow an ER mechanism** [29]. This suggests that the production of DMC from CO₂ instead of CO may take a very different pathway for the production of DMC despite the apparent similarity of the overall reaction.”

Changes of Reference:

A new reference was introduced as Ref 25 with Eley–Rideal Mechanism. The old “Ref 25” was shifted to [26], for rate law derivations.

25. Eta V, Maki-Arvela P, Warna J, Salmi T, Mikkola JP, Murzin DY. 2011 Kinetics of dimethyl carbonate synthesis from methanol and carbon dioxide over ZrO₂–MgO catalyst in the presence of butylene oxide as additive *Applied Catalysis A*. **404**, 39-46. (doi:10.1016/j.apcata.2011.07.004).

26. Martin CM, Li L, Bhalkikar A, Doyle JE, Zeng XC, Cheung CL. 2016 Kinetic and mechanistic investigations of the direct synthesis of dimethyl carbonate from carbon dioxide over ceria nanorod catalysts, *Journal of Catalysis*. **340**, 295-301. (doi:10.1016/j.jcat.2016.06.003)

The reason we cite original Ref. [25] Martin et al. (2016):

We cited original [Ref. 25] because we focused on Derivations of apparent rate laws in Supplementary Information instead of their results and conclusion.

Supplementary Information:

<https://www.sciencedirect.com/science/article/pii/S0021951716300835?via%3Dihub>

Honestly, **the methodology of derivations of apparent rate laws** were important. We emphasize on **the development of elementary steps and derivation of rate laws and rate limiting step.**

(1) The valuable results from Ref. 25: Fig 8 of Ref. [ER] and Supplementary Information.

Fig. 1 was generated based on the ER mechanism of Figure 8, as a start-up.

Figure 8 with [ER model] the reference was added into manuscript.

[ER] Eta V, Maki-Arvela P, Warna J, Salmi T, Mikkola JP, Murzin DY. 2011 *Applied Catalysis A*. **404**, 39-46. (doi:10.1016/j.apcata.2011.07.004).

Fig.1 Mechanisms of Catalytic carbamate formation based on Eley-Rideal model

Moreover, Fig. 8 included “Elementary mechanistic steps proposed for the formation of DMC from CO₂ and methanol by a Langmuir–Hinshelwood and an Eley–Rideal mechanism [16,18].” Apparent rate law was shown for each step if the given step was significantly slower than the preceding step(s).

Honestly, the original ref. [25] was **NOT** cited for the final results and conclusions. **Their methodology of derivations of apparent rate laws** was valuable. We emphasize on **the development of elementary steps and derivation of rate laws and rate limiting/determining steps.**

From the **Supplementary Information with rate law derivations** of Reference:

“Derivations of Apparent Rate Laws Shown in Figure 8:

A. Eley-Rideal Mechanism Elementary Steps from Figure 8

- 1) $\text{MeOH} + * \rightleftharpoons \text{MeOH}^*$
- 2) $\text{MeOH}^* + \text{CO}_2 \rightleftharpoons \text{MC}^*$
- 3) $\text{MC}^* + \text{MeOH}^* \rightleftharpoons \text{DMC} + \text{H}_2\text{O} + *$

If step 1 is rate limiting

B. Langmuir-Hinshelwood Mechanism Elementary Steps from Figure 8

- 1) $\text{CO}_2 + * \rightleftharpoons \text{CO}_2^*$
- 2) $\text{MeOH} + * \rightleftharpoons \text{MeOH}^*$

- 3) $2\text{MeOH}^* + \text{CO}_2^* \rightleftharpoons \text{DMC}^* + \text{H}_2\text{O}^* + *$
- 4) $\text{DMC}^* \rightleftharpoons \text{DMC} + *$
- 5) $\text{H}_2\text{O}^* \rightleftharpoons \text{H}_2\text{O} + *$

If step 1 is rate limiting

From our study, **different reactions and catalysts** would result in different mechanisms. The experimental results would verify different rate models finally. That was the main reason that our model was ER mechanism, different from their model of LH mechanism.

In conclusion, **the methodology through which they developed the “Rate law if rate determining” was the foundation of our study**, which was based on steady state approximation. We developed our “rate law if rate determining” based on their methodology.

(2) The major difference from their results and our study under ER mechanism.

Even under the Eley-Rideal mechanism that they tried to develop, the elementary steps were different: 3 steps for them and 4 steps for us. Different elementary steps would derivate different rate limiting steps as follows:

Eley-Rideal Mechanism Elementary Steps from Synthesis of DMC [25].

- 1) $\text{MeOH} + * \rightleftharpoons \text{MeOH}^*$
- 2) $\text{MeOH}^* + \text{CO}_2 \rightleftharpoons \text{MC}^*$
- 3) $\text{MC}^* + \text{MeOH}^* \rightleftharpoons \text{DMC} + \text{H}_2\text{O} + *$

Elementary reaction step	Rate law if rate determining
$\text{MeOH} + * \rightleftharpoons \text{MeOH}^*$	$R = k [\text{MeOH}] [*]$
$\text{MeOH}^* + \text{CO}_2 \rightleftharpoons \text{MC}^*$	$R = k [\text{MeOH}] [\text{CO}_2] [*]$
$\text{MC}^* + \text{MeOH}^* \rightleftharpoons \text{DMC} + \text{H}_2\text{O} + *$	$R = k [\text{CO}_2] [\text{MeOH}]^2 [*]^2$

Eley-Rideal Mechanism Elementary Steps from carbamate formation reactions. (Our Study)

- 1) $\text{RNH}_2 + * \rightleftharpoons \text{RNH}_2^*$
- 2) $\text{RNH}_2^* + \text{CO}_2 \rightleftharpoons \text{Zwitterion}^*$
- 3) $\text{Zwitterion}^* + \text{H}_2\text{O} \rightleftharpoons \text{Carbamate}^* + \text{H}_3\text{O}^+$
- 4) $\text{Carbamate}^* \rightleftharpoons \text{Carbamate} + *$

Table 2, with the developed rate limiting steps (Part)

RDS	Rate law if rate determining, with full and simplified formats
B1	$r_1 = K_B \left\{ \frac{[A] - K_A \frac{[C][H^+]}{p_B}}{1 + K_4[C] + K_3 K_4 [C][H^+] + \frac{K_2 K_3 K_4 [C][H^+]}{p_B}} \right\} \quad r_1 = K_B \left\{ \frac{[A]}{1 + K_4[C]} \right\} \text{ (simple)}$

$$\begin{aligned}
\text{B2} \quad r_2 &= K_B \left\{ \frac{[A]p_B - K_1K_2K_3K_4[C][H^+]}{K_1 + K_1K_4[C] + K_1K_3K_4[C][H^+] + [A]} \right\} \quad r_2 = K_B \left\{ \frac{[A]p_B}{K_1 + K_1K_4[C] + [A]} \right\} \text{ (simple)} \\
\text{B3} \quad r_3 &= K_B \left\{ \frac{K_1K_2[A]p_B - K_3K_4[C][H^+]}{1 + K_4[C] + K_1K_2[A]p_B + \frac{[A]}{K_1}} \right\} \quad r_3 = K_B \left\{ \frac{K_1K_2[A]p_B}{1 + K_4[C] + K_1K_2[A]p_B + \frac{[A]}{K_1}} \right\} \text{ (simple)} \\
\text{B4} \quad r_4 &= K_B \left\{ \frac{\frac{K_1K_2}{K_3}[A]p_B - K_3K_4[C][H^+]}{[H^+] + \frac{K_1K_2[A]p_B}{K_3} + K_1K_2[A][H^+]p_B + \frac{[A]}{K_1}[H^+]}} \right\} \quad r_4 = K_B \left\{ \frac{\frac{K_1K_2}{K_3}[A]p_B}{\frac{K_1K_2[A]p_B}{K_3}} \right\} = K_B \text{ (simple)}
\end{aligned}$$

(3) The main reason that ER mechanism are more suitable for our study

The biggest difference was the acid base properties of our study. For their studies under the Langmuir-Hinshelwood mechanism, CO₂ was acidic gas and MeOH steam was neutral. Both molecules could be adsorbed onto the surface of ceria catalysts together via LH model: reactants A and B were adsorbed onto solid surface first, and then reacted with each other.

However, for our case, CO₂ is acidic but MEA (RNH₂) consists of basic molecules. CaCO₃, MgCO₃ and BaCO₃ contain abundant basic active sites on the surface, and they were pre-absorbed with MEA molecules. CO₂ would instantly react with the MEA on the solid surface when the molecule was approaching the solid surface rather than the adsorbed onto the solid surface. If CO₂ was adsorbed onto the solid surface without colliding any amine molecules, the carbamate formation did not occur at all. The CO₂ + MEA reaction on solid surface was more suitable for Eley-Rideal mechanism (ER): a gas-phase reagent CO₂ directly reacted with an adsorbed species (adspecies) of RR'NH.

Changes:

Several place were also corrected in text, that the ER model was applicable with CO₂ + RR'NH with catalyst MCO₃. (This model was applicable under specific catalytic cases)

Section 1. Section 2.2 1st paragraph was modified.

As mentioned, both Eley-Rideal model and Langmuir-Hinshelwood model mechanisms were applicable in the field of heterogeneous catalytic reactions on gas-solid interface [26]. After detailed investigation and analysis, Eley-Rideal model could be more suitable for carbamate formation reaction because of the acid-base nature of CO₂ and RNH₂. CaCO₃, MgCO₃ and BaCO₃ contained abundant basic active sites on the surface, and it was pre-absorbed with MEA molecules from experimental procedures. From Figure 1, the CO₂ molecules reacted with the MEA that was instantaneously adsorbed on the solid surface when approaching the surface. CO₂ was unlikely to be directly adsorbed onto the solid surface. If CO₂ was adsorbed onto the solid surface without collision of any amine molecules, the carbamate formation reaction did not occur at all. In the Eley-Rideal mechanism, a gas-phase reagent

such as CO₂ directly reacted with an adsorbed species (adspecies) RR'NH, and the product carbamate either desorbed or remained adsorbed on the surface depending on the exothermic reaction [24]. Mostly, the product desorbed the surface due to the heat release. [24]

Question 3. The term of liquid [OH⁻]

3. The manuscript mentioned a lot of unusual terms. The reviewer does not understand the meaning of these terms. For example, what is liquid [OH⁻]?

Response:

The reviewer's suggestion was well taken. Some terms has been revised. The term "liquid [OH⁻]" has been revised as "[OH⁻] anions in liquid phase" as emphasis on homogeneous catalysis.

Changes:

The term of "liquid [OH⁻]" has been revised as "[OH⁻] anions in liquid phase".

Question 4. The term of $r_{OH^-}^*$ and $k_{OH^-}^*$,

4. Many of the mathematics symbols were not defined in the main text. It is difficult to follow the discussions. For example, on page 5, line 24, what are k (superscript: dot) (subscript: OH⁻) and r (superscript: dot) (subscript: OH⁻)?

Response:

The reviewer's suggestion was well taken. The introduction of kinetics studies required expansion with more details within reaction mechanism (Section 2.1). To be honest, these mathematic symbols were not defined by us, and all of them were quoted directly from references. [4-21] We only placed the most relevant equations in the theory among hundreds of mathematic equations. Some backgrounds were omitted to avoid the duplications.

After we carefully reviewed the references of kinetic studies, **the symbol should be $r_{OH^-}^*$ and $k_{OH^-}^*$, and it should be a star "*" rather than dot "."**[2, 21].

These symbols represented the reaction rates and rate constant of bicarbonate formation [6, 21]:

Bicarbonate formation:

$$r_{OH^-}^* = k_{OH^-}^* [OH^-] [CO_2]; \quad k_{OH^-}^* = 8322 \text{ m}^3/\text{kmol at 298K} \quad [21]$$

The main reactions that occurred in the MEA-CO₂-H₂O system for kinetics study was listed below [2,6], and Blauwhoff et al.(1983)[6] conducted the kinetics of CO₂-Amine in aqueous solutions and generated the main equations.

Bicarbonate formation $\text{CO}_2 + \text{OH}^- \rightleftharpoons \text{HCO}_3^-$ [2,6]

Then, the overall reaction rate could be written as below, consisting of two parts:

$$\begin{aligned} r_{\text{ov}} &= r_{\text{CO}_2-\text{R}_1\text{R}_2\text{NH}} + r_{\text{OH}^-}^* = k_{\text{ov}}[\text{CO}_2] \\ &= k_{\text{app}}[\text{Amine}][\text{CO}_2] + k_{\text{OH}^-}^*[\text{OH}^-][\text{CO}_2] \quad [2,6] \end{aligned}$$

Furthermore, Blauwhoff et al.(1983)[6] expanded the rate equations along with the rate constants in detail. The overall rate constant k_{ov} covers the contributions of both reactions as below:

$$k_{\text{ov}} = k_{\text{OH}^-}^*[\text{OH}^-] + \frac{k_{2,\text{R}_1\text{R}_2\text{NH}}[\text{R}_1\text{R}_2\text{NH}][\text{CO}_2]}{1 + \frac{k_{-1}}{k_{2,\text{R}_1\text{R}_2\text{NH}}[\text{R}_1\text{R}_2\text{NH}] + k_{\text{OH}^-}[\text{OH}^-] + k_{\text{H}_2\text{O}}[\text{H}_2\text{O}]}}$$

References [2, 6, 21] in Manuscript:

2. Sema T, Naami A, Liang ZW. 2012 Solvent chemistry: reaction kinetics of CO₂ absorption into reactive amine solutions. Carbon Manag. 3,201-220. (doi: 10.4155/cmt.12.13)

6. Blauwhoff PMM, Versteeg GF, Swaaij WPM. 1983 A study on the reaction between CO₂ and alkanolamines in aqueous solutions. Chem. Eng. Sci. 38, 1411-1429.(doi:10.1016/0009-2509(83)80077-3)

21. Versteeg GF, Van Dijk LAJ, Van Swaaij WPM. 1996 On the kinetics between CO₂ and alkanolamines both in aqueous and non-aqueous solutions, Chem. Eng. Commu. 144, 113-158. (doi: 10.1080/00986449608936450)

Changes: 1st paragraph added below the title of 2.1.

The main reactions were listed below of CO₂ reaction with primary/secondary amines in aqueous solutions firstly [2,6] Blauwhoff et al.(1983)[6] has already generated the main equations after conducting the kinetics of CO₂-Amine in aqueous solutions.

Carbamate formation $\text{CO}_2 + 2 \text{R}_1\text{R}_2\text{NH} \rightleftharpoons \text{R}_1\text{R}_2\text{N-COO}^- + \text{R}_1\text{R}_2\text{NH}^+$ (1)[2, 6]

Bicarbonate formation $\text{CO}_2 + \text{OH}^- \rightleftharpoons \text{HCO}_3^-$ (2) [2, 6]

Then, the overall reaction rate consists of two parts below:

$$r_{\text{ov}} = r_{\text{CO}_2-\text{R}_1\text{R}_2\text{NH}} + r_{\text{OH}^-}^* = k_{\text{ov}}[\text{CO}_2] = k_{\text{app}}[\text{Amine}][\text{CO}_2] + k_{\text{OH}^-}^*[\text{OH}^-][\text{CO}_2] \quad (3) [2,6]$$

Furthermore, Blauwhoff et al. [6] developed the rate constants in detail. The overall rate constant k_{ov} covers the contributions of both reactions and can be written as below:

$$k_{\text{ov}} = k_{\text{OH}^-}^*[\text{OH}^-] + \frac{k_{2,\text{R}_1\text{R}_2\text{NH}}[\text{R}_1\text{R}_2\text{NH}][\text{CO}_2]}{1 + \frac{k_{-1}}{k_{2,\text{R}_1\text{R}_2\text{NH}}[\text{R}_1\text{R}_2\text{NH}] + k_{\text{OH}^-}[\text{OH}^-] + k_{\text{H}_2\text{O}}[\text{H}_2\text{O}]}} \quad (4) [6]$$

The bicarbonate formation was not dominant if compared with carbamate formation for condensed amine solutions. [2,6] The rate equation $r_{\text{OH}^-}^*$ and constant $k_{\text{OH}^-}^*$ has already been developed [2, 21].

$$r_{\text{OH}^-}^* = k_{\text{OH}^-}^*[\text{OH}^-][\text{CO}_2] \quad (5) [2, 6, 21]$$

$$k_{OH^-}^* = 8322 \text{ m}^3/\text{kmol at 298K} \quad (6) [21]$$

The focus of this study was on “carbamate formation”, in terms of reaction rate $r_{CO_2-R_1R_2NH}$ and rate constant k_{app} . They were much bigger than $r_{OH^-}^*$ and $k_{OH^-}^*$. [2]

Question 5. The reason to select BaCO₃

5. Why was BaCO₃ chosen for the evaluation?

Response:

The reviewer has proposed a good question. We selected three types of metal carbonates, CaCO₃, MgCO₃ and BaCO₃ as a group. They belonged to alkaline earth metals (Be, Mg, Ca, Sr, Ba, etc) of IIA group in the Periodic Table.

The kinetic model of this study was not only applicable to one metal carbonate, but also to a group of metal carbonates from Periodic Table. Moreover, among the data base of 39 sets of (X_A, t), 12 sets were for CaCO₃, 12 sets were for MgCO₃ and 15 sets were for BaCO₃. If without BaCO₃, there were only 24 sets.

Changes: Introduction Final paragraph

Within this study, surface of solid chemicals. “The CaCO₃, MgCO₃ and BaCO₃ were selected as a group, for the metals were alkaline-earth metals belonging to IIA group in the Periodic Table.”

Question 6. The title of Figure 1

6. Page 7, line 45, what is figure 1a?

Response:

There was only one figure for Figure 1. This “a” was the footnote of explanation of Title of Table 2. “Table 2. The elementary reaction steps and rate law if rate determining of Fig 1.”

Changes: The title of Table 2 was changed, and the “a” was put into Row 1.

Table 2. The elementary reaction steps and rate law if rate determining of Fig 1

No.	Elementary reaction steps of catalytic carbamate formation	Rate constant
	Eley–Rideal model ^a	

Question 7. The steady state approximation

7. It is unclear to the reviewer why r₃ is zero for B₃ when B₁ is the rate determining step. According to the elementary step, B₃ is a one direction elementary step reaction.

Response:

The reviewer has raised a good question. The rate models based on “Steady-State approximation” as a classical method in the kinetics of reaction rates. If B₁ was the rate determining step, r₁ ≠ 0, and the rates of r₂, r₃ and r₄ were very small and set to 0.

$r_1 \neq 0, r_2 = r_3 = r_4 = 0$. “Steady-State approximation”.

Many other researchers developed the Apparent rate law with this approximation. [25, 26]

The “Steady-State approximation” was introduced. Below was its definition in Wikipedia.

[https://en.wikipedia.org/wiki/Steady_state_\(chemistry\)#Reaction_rates](https://en.wikipedia.org/wiki/Steady_state_(chemistry)#Reaction_rates)

“Steady state approximation in chemical kinetics. The steady state approximation, occasionally called the stationary-state approximation, **involves setting the rate of change of a reaction intermediate in a reaction mechanism equal to zero** so that the kinetic equations can be simplified by setting the rate of formation of the intermediate equal to the rate of its destruction.

In practice, it is sufficient that the **rates of formation and destruction are approximately equal, which means that the net rate of variation of the concentration of the intermediate is negligible** compared to the formation and destruction, and the concentration of the intermediate varies slowly.”

Therefore, the introduction of steady-state approximation was explained in Theory.

Changes: Section 2.3 1st paragraph

“It was originated from the steady-state approximation, a classical method in the kinetics of reaction rates. [25-27] The steady-state approximation involved setting the rate of change of a reaction intermediate in a reaction mechanism equal to zero, and then kinetic equations could be simplified. For instance, if B1 assumed to be the rate-determining step, the rest 3 steps were regarded as steady-state, and the rate of formation of the intermediate equaled to the rate of its destruction, so that the overall apparent rates r_2 to r_4 were set to 0.”

Question 8. The typos

8. There are many typos in the manuscript. For example, “concertation” on page 9, line 16.

Response:

The reviewer’s suggestion has been well adopted and the whole manuscript was carefully proofread to remove these typos.

Question 9. The equations of X_A vs α

9. There is an error for equation (12). It should be $[C] = CA_0 \text{ times } \alpha = \frac{1}{2} CA_0 (1 - X_A)$. The manuscript lacks clear derivations of the rate equations.

Response:

The reviewer’s suggestion is honored. We reviewed the equation (10-12) carefully, and we discovered that **Eqn(10) is wrong and Eqn(12) is correct**. The revised equations should be:

$$X_A = 2 \times \alpha \quad (10) \quad (\alpha < 0.40; X_A < 0.80)$$

$$[\text{Amine}] = C_{A0} (1 - 2 \times \alpha) = C_{A0} (1 - X_A) \quad (11)$$

$$[\text{Carbamate}] = C_{A0} \times \alpha = \frac{1}{2} C_{A0} X_A \quad (12)$$

The equations were developed from carbamate formation reaction at CO₂ loading < 0.40 mol/mol:

C_{A0}	0	0	Initial
$C_{A0} (1-2\alpha)$	$C_{A0} \alpha$	$C_{A0} \alpha$	α loading
$C_{A0} (1-X_A)$ (free amine)	$\frac{1}{2}C_{A0}X_A$	$\frac{1}{2}C_{A0}X_A$	X_A conversion

Therefore, $X_A = 2 \times \alpha$.

The conversion was two times of CO₂ loading based on the stoichiometric ratio of 2:1 of eqn (1).

For MEA, this equation was valid at range of 0-0.40 mol/mol based on ion speciation plot.

Moreover, [A] represented the concentration of free amine, and it was modified as [Amine]. [C] represented the concentration of carbamate and it was modified as [C_{Carbamate}].

Changes: Eqs (10–12) was revised and re-numbered to (12–14) as below:

$$X_A = 2 \times \alpha \quad (12) \quad (\alpha < 0.40; X_A < 0.80)$$

$$[\text{Amine}] = C_{A0} (1 - 2 \times \alpha) = C_{A0} (1 - X_A) \quad (13)$$

$$[\text{Carbamate}] = C_{A0} \times \alpha = \frac{1}{2} C_{A0} X_A \quad (14)$$

Question 10. The experimental procedures

10. The manuscript lacks detailed experimental procedures for reproducing the materials.

Response:

The reviewer's suggestion is well taken, and the detailed procedures was provided into the manuscript. The experimental procedure of CO₂ absorption with MEA was similar to other publication [23], which is CO₂ adsorption with DEA with CaCO₃ and MgCO₃. This question is similar to Q1, and we solve them together.

Changes: Section 3.1 added paragraph 2, (Same as Question 1)

Question 11. The criteria of data selection with R² > 0.99

11. The authors should justify why they chose R² > 0.990 as the criteria for good fittings of the data.

Response:

The reviewer's suggestion was taken, some backgrounds need to be explained.

The heterogeneous catalysis of CO₂ reaction with R₁R₂NH has been recently discovered since 2013. Several publications has already verified the catalysis with experimental results, [23,30] but there are very little precedent studies of mechanistic models. The development of elementary steps with derivation of apparent rate law model was the same as reference [25, 26], and presented in Table 2.

Table 2, with the developed rate limiting steps (Part)

RDS	Rate law if rate determining, with full and simplified formats	
B1	$r_1 = K_B \left\{ \frac{[A] - K_A \frac{[C][H^+]}{p_B}}{1 + K_4[C] + K_3K_4[C][H^+] + \frac{K_2K_3K_4[C][H^+]}{p_B}} \right\}$	$r_1 = K_B \left\{ \frac{[A]}{1 + K_4[C]} \right\}$ (simple)
B2	$r_2 = K_B \left\{ \frac{[A]p_B - K_1K_2K_3K_4[C][H^+]}{K_1 + K_1K_4[C] + K_1K_3K_4[C][H^+] + [A]} \right\}$	$r_2 = K_B \left\{ \frac{[A]p_B}{K_1 + K_1K_4[C] + [A]} \right\}$ (simple)

However, it was also difficult to verify the kinetic mechanism with B1 or B2, because most reactions occurred on the solid surface and the parameters of K₁-K₄ were also unknown. We have no choice but to verify the rate model from the fundamental of reaction kinetics: the definition of reaction rate “ $r = dC_A/dt$ ”. [27] Based on the Textbook of “Chemical Reaction Engineering”, the method was “Integrated analysis”. [27] This method has been rarely adopted since most kinetic models were verified based on precedent publications.

Integrated Analysis of rate models with selection criteria:

The definition of reaction rates is “ $r = -dC_A/dt = C_{A0} \frac{dX_A}{dt}$ (15)”. It was integrated and adopted $C_A = C_{A0}(1-X_A)$ with different formats of rate equations (r) at the left side. We established the direct correlation of “ $r = C_{A0} \frac{dX_A}{dt}$ ” with its integrated format of “ $F(X_A) = Kt$ ”. We developed several different integrated rate equations of $F(X_A) = Kt$ representing different rate equations (r) from (B1) and (B2) under several sub-cases. There was only one set of dataset of (X_A , t) that were obtained from experiments. However, there were several different formats of $F(X_A)$ based on different rate equations (r) in Table 2. **Consequently, there was one accurate model fitting data (X_A , t) among Eq(16-18) including different subcases.**

These integrated rate equations (16-18) were developed with a general format of “ $F(X_A) = K t$ ”, which was a **linear** equation of “ $y = k x$ ”. For each set of data ($F(X_A)$, t) based on (X_A , t), there were 5 points on one curve with different conversion of $X_A = 0.0, 0.4, 0.5, 0.6, 0.7$ and 0.8 . Different curves of ($F(X_A)$, t) reflect different r .

If the curve of “ $F(X_A) = Kt$ ” was linear and straight, it meant ($F(X_A)$, t) was fitting for the model

and its specific rate equation (r) was also fitting for the mechanism. The higher R^2 of the lines of $F(X_A) = Kt$ indicated the better fitting of the dataset of “ $r = C_{A0} \frac{dX_A}{dt}$ ” responds to experimental dataset of (X_A, t) . Therefore, we selected the highest standard $R^2 > 0.99$ as the criteria.

Changes: Section 3.3, was totally modified as below.

There was little precedent research of rate model verification of the developed equations of B1-B4 in Table 2 and the parameters of K1-Kx were hard to calculate, so that the rate equations needed to be verified from the definition of reaction rates $r_A = -\frac{dC_A}{dt}$ as the fundamental of reaction kinetics.[27] With database of (X_A, t) in Table SA.1-3, “Integral Method of Analysis” was adopted for rate equation validation[7], which was a fundamental methodology of chemical reaction engineering.[27]

The goal was to establish the direct correlation of “ $r = C_{A0} \frac{dX_A}{dt}$ (15)” with its integrated format of “ $F(X_A) = Kt$ ”. We developed several proper integrated equations of $F(X_A) = Kt$ with the combination of the definition of reaction rate of Eq(15) and the specific format of reaction rate (r) of Eqn(16-18) in order to verify B1 or B2 as the rate determining step. The $[CO_2]$ was not included, for it had already been verified as the 1st order for absorption.[2]

The rate equation of CO_2 absorption was Eq(15) by amine concentration (C_A) or conversion (X_A) at the right side:

$$r_A = -\frac{dC_A}{dt} = -\frac{dC_{A0}(1-X_A)}{dt} = \frac{C_{A0}dX_A}{dt} \quad (15)$$

For the left side, $r_A = f(X_A)$ was adopted by either B1 or B2 in Table 2. The integrated rate equations were carried-out with Eq(16-18) under different sub-cases with brief analyses in Appendices. There was only one set of dataset of (X_A, t) obtained from experiments. However, there were several different formats of $F(X_A)$ Eq(16-18) based on different rate equations (r) in Table 2. Consequently, there was one accurate model fitting data (X_A, t) among Eq(16-18), different subcases included.

$$\text{RDS} = \text{B1:} \quad r_1 = K_B \left\{ \frac{[A]}{1 + K_4[C]} \right\}$$

$$\ln \frac{1}{1-X_A} = K_B t \quad r_1 = K_B [A] \quad (16) \quad \frac{\theta_{\text{carbamate}}}{\theta_{\text{empty}}} \ll 1$$

$$\ln \frac{1}{1-X_A} - X_A = K' t \quad r_1 = K_B \frac{[A]}{K_4[C]} \quad (17) \quad \frac{\theta_{\text{carbamate}}}{\theta_{\text{empty}}} \gg 1$$

$$\text{RDS} = \text{B2:} \quad r_2 = K_B \left\{ \frac{[A]p_B}{K_1 + K_1K_4[C] + [A]} \right\}$$

$$k_a \left\{ \ln \frac{1}{1-X_A} - X_A \right\} + X_A = K'' t \quad (18)$$

$$ka = 1/2 K_1 K_4 = 0.05, 0.005, \text{ and } 0 \text{ for sub-cases}$$

For Eq(18), ka reflected the ratio of $\frac{\theta_{\text{carbamate}}}{n_{\text{sites}}}$. To verify the rate models of RDS = B2, ka was selected by 0.05, 0.025 and 0.005, representing that 10%, 5% and 1% of the total active sites were covered with carbamate. The extreme condition was also tested, where $ka = 0$ (similar to 0th order).

Each sub-case of Eqs(16-18) was verified by 39 sets of experimental data of (X_A, t) . Each set of (X_A, t) with its specific parameter (concentration, catalyst, mass i.e. 1.0 mol/L, CaCO₃, 5g) generated different sets of $(F(X_A), t)$ based on different RDS of Eq (16-18). Different curves of $(F(X_A), t)$ reflected different rate equations (r). For each set of data $(F(X_A), t)$, there were 5 points with different conversion of $X_A = 0.0, 0.4, 0.5, 0.6, 0.7$ and 0.8 . These curves of $(F(X_A), t)$ could be either linear or curvy.

These integrated rate equations (16-18) were developed with a general format of “ $F(X_A) = K t$ ”, which was a **linear** equation of “ $Y = k X$ ”. If the curves of $(F(X_A), t)$ were linear and straight, it meant $(F(X_A), t)$ was quite fitting for “ $F(X_A) = K t$ ” of Eqs(16-18), and its related specific rate equation (r) was fitting for the mechanism of B1 or B2. The higher R^2 of the lines of “ $F(X_A) = Kt$ ” of Eqs(16-18) indicated the better fitting of the dataset of “ $r = C_{A0} \frac{dX_A}{dt}$ ” responded to experimental dataset (X_A, t) . Therefore, we selected the highest standard $R^2 > 0.99$ as the criteria. After repeated verifications of 39 sets of experimental data (X_A, t) with-linear regressions, the set containing most lines of $R^2 > 0.99$ of Eqs(16-18) were verified as the highly accurate rate model.

Question 12. The characterization data for BaCO₃.

12. Why did the authors present no characterization data for BaCO₃ ?

Response :

The reviewer’s raised a good question.

We conducted SEM of BaCO₃ under 20μm and add into Fig 2.

SEM of BaCO₃ 20μm

For BET surface area of BaCO₃, a recent study (Nov 2018) published the Characterization of BaCO₃, the BET surface area as 4.66 m²/g with pore volume of 0.008 cm³/g, and pore size of 6.46 nm. (New Ref 29: Afari et al. *Ind. Eng. Chem. Res.* 2018, 57, 15824–15839).

S3. SEM images of all solid base catalysts studied (a) BaCO₃ (b) CaCO₃ (c) Ca(OH)₂ (d) Hydrotalcite (e) Cs₂O/Al₂O₃ (f) Cs₂O/α-Al₂O₃ (g) K/MgO

(a) BaCO₃, (b) CaCO₃ at 100μm in (Support information) with link below

https://pubs.acs.org/ccindex.cn/doi/suppl/10.1021/acs.iecr.8b02931/suppl_file/ie8b02931_si_001.pdf

Table 4. Physical and Chemical Properties of Catalysts Studied

catalyst	BET surface area (m ² /g)	pore volume (cm ³ /g)	pore size (nm)	number of basic sites ^a (a.u)	basic strength (°C)
K/MgO	63.33	0.270	17.08	58785	237
Ca(OH) ₂	8.43	0.042	19.89368	15565	862
Cs ₂ O/α-Al ₂ O ₃	7.61	0.041	21.44	8619	132
BaCO ₃	4.66	0.008	6.46		
Cs ₂ O/Al ₂ O ₃	3.33	0.009	10.33	818	555
hydrotalcite	0.46	0.0001	0.86	28258	179
CaCO ₃					

^aArbitrary units.

Ref [29]: Afari, D.B., Coker, J., Narku-Tetteh, J., Idem, R. **2018** Comparative kinetic studies of solid absorber catalyst (K/MgO) and solid desorber catalyst (HZSM-5)-aided CO₂ absorption and desorption from aqueous solutions of MEA and blended solutions of BEA-AMP and MEA-MDEA, *Ind. Eng. Chem. Res.*, **57**(46), 15824-15839. (DOI: 10.1021/acs.iecr.8b02931)

Changes:

The SEM of BaCO₃ was added into Figure 2.

The data of BET surface area was also provided into manuscript.

Question 13. The errors of experimental data

13. What are the error bars for the experimental data?

Response:

The reviewer has mentioned a very good question. Our dataset of (X_A, t) was calculated based on the experimental data of CO₂ loading of (α, t) of CO₂ absorption profile.

The CO₂ loading (α) of the amine solution was tested for different amine samples (1ml each time)

from a set of Chittick apparatus, and the AAD% of the experimental tests was about 2.5%. [23] The error and deviation were repeatedly reported.

CO₂ loading was calculated based on the equations at 298K:

$$\alpha = \frac{(V_{total} - V_{HCl})}{C_{amine} V_{amine} 24.0 \times 10^{-3}} \frac{\times 10^{-3} L}{\frac{mol\ amine\ L}{mol\ CO_2}} \text{ mol CO}_2/\text{mol amine}$$

Changes: Section 3.2 end of first paragraph

The absorption profile of CO₂ loading (α) vs time were completed elsewhere. The CO₂ loading (α) of the amine solutions was tested from Chittick apparatus, and the AAD% of the experimental tests was 2.5%. [23] The conversion of amines X_A was calculated from CO₂ loading (α) with equations below.

Question 14. Our method of F(XA) vs time vs power law model of rate order

14. It is difficult to evaluate the rate law by fitting the F(Xa) vs time. It is better to directly extract the order of reactions with respect to each reactants from the experimental data.

Response:

The reviewer has raised a very important point, and it has to be discussed in detail. We have already extracted the order of reactions with [MEA] from the experimental data right after the experiments. The rate order was calculated and discussed at the last part of R&D. (Check Summary of this manuscript below). The reaction rate was the 1st order for both catalytic and non-catalytic absorptions for [MEA]. The way of presenting rate order was the same as others [26], which **presented rate order at the end of R&D.**

The method quoted from the reviewer was the “Power Law model” with a graphic method of kinetics, but it was an **over-simplified model** lacking important information related to intrinsic mechanism.

(1) The results of our study related to rate order:

The verified rate equation of our study was RDS = B1, with equation (B1) based on the experimental data. Desorption of Carbamate resulted in the very low value of K₄[C], and then the rate equation was simplified again to $r_1 = K_B[A]$ (B5), which was the **pseudo-first order for MEA.**

$$r_1 = K_B \left\{ \frac{[A]}{1 + K_4[C]} \right\}; \quad (B1)$$

$$K_4[C] = \frac{\theta_{carbamate}}{\theta_{empty}} \ll 1; \text{ abundant empty active sites, few carbamate remained adsorbed}$$

$$r_1 = K_B[A] \quad \text{Pseudo 1}^{\text{st}} \text{ order for [MEA]} \quad (B5)$$

Even the reaction order was the 1st order of (B5), the same for catalytic reaction and non-catalytic reaction, the rate model and intrinsic mechanism were different. The (B5) was a simplified rate equation, and our study revealed much important information besides rate order and rate constant.

Summary of our Study had been provided in **Cover letter:**

“This work involved a mechanistic and kinetic study of a specific heterogeneous catalytic reaction of CO₂ + primary/secondary amines (RR’NH) via carbamate (RR’N-COO⁻) formation. (1) A new mechanism was proposed as the Eley-Rideal model. (2) Four elementary reaction steps were developed based on the mechanism. (3) The rate laws if rate determining of these 4 steps were developed with full and simplified formats. (4) The rate determining step of the special case of MEA was determined as the 1st step: amine adsorption onto the solid surface. (5) The rate order was pseudo-1st order for MEA, and the enhancement of solid chemicals was 1.2-2.0 times higher than the 2nd rate constant “kz” with existence of different masses and types of CaCO₃, MgCO₃ and BaCO₃.”

(2) The power law model among the methods of kinetic study development:

Based on a review of kinetic study of non-catalytic CO₂-amine reactions [2]. The absorption rate data of CO₂ absorption equipment were interpreted by means of **Stage (1) graphical methods such as Power Law Model**, **Stage (2) simplified kinetics models based on reaction mechanisms** (i.e., zwitterion, termolecular or base-catalyzed hydration mechanisms), and **Stage (3) comprehensive numerically solved reaction kinetics models** that coupled with chemical equilibrium, mass-transfer and kinetics of all possible reactions.

The “Power Law Model” was a straightforward but over-simplified one. This graphical method was used from 1980s to mid-1990s to determine the order of reaction, and the reaction rate constant of MEA, DEA, TEA and AMP. [2] Generally, the reaction order with respect to [CO₂] was the 1st for all aqueous–amine solutions. [2] The power law model could extract the rate order of the reactions directly, but it could not provide detailed intrinsic reaction mechanism or detailed information of the kinetics, such as elementary steps, rate determining steps, the molecular interactions, etc.

Based on the review of kinetics [2], the Power Law Model (1) was not better than kinetics models based on reaction mechanisms (2) which involved verification of the apparent rate law by fitting the F(X_A) vs time.

(3) The purpose of using $F(X_A) = K t$ is for rate model verification

Based on **Appendices**, there were several rate equations corresponding to different RDS with B1

(Amine adsorption) or B2 (Zwitterion formation with N-C bond formation) under different sub-cases. Each case should be equally treated to special amine MEA.

The fitting $F(X_A)$ vs time was conducted in order to verify the most suitable rate equations among different RDS with corresponding experimental dataset (X_A , t). This process was time consuming, but it was necessary to verify the correct rate-determining steps (RDS) for special cases of MEA to reveal the reaction mechanism on the solid surface. The rate model verification with experiment data by fitting $F(X_A)$ vs time was very important.

Changes: Section 4.3, 3rd Paragraph

Based on Eq(16), $r_1 = K_B[A]$. This equation also indicated that the heterogeneous catalytic CO₂ absorption was the pseudo 1st order with respect to [MEA], the same as non-catalytic absorption. The rate of heterogeneous catalytic CO₂-MEA absorption was Eq(19), similar to Eq(4) of non-catalytic absorption. Despite the same reaction orders, the mechanisms and rate constants were different under catalytic and non-catalytic absorptions. The reaction order could also be directly extracted from graphical method based on the experimental data [2], but the Power Law model was a straightforward but over-simplified method lacking detailed intrinsic reaction mechanism and elementary steps.[2] The developing rate law based on the definition of reaction rate was hard and time-consuming, but it accurately verified the mechanism and elementary steps.

Reviewer 2:

Comments to the Author(s)

This manuscript proposed a new model for heterogeneous catalytic CO₂-RR' NH interactions. The reaction process, the elementary steps, and rate limiting steps was developed. These results are useful for further studies with secondary amines etc. This manuscript is worth publication.

Several small problems need to be edited to improve the quality:

Question 1. The kinetic of R₃N in introduction.

1. Introduction:

(1) It is mentioned mostly that the reaction of CO₂ with primary amine and secondary amine, but little of CO₂ with tertiary amine. Please mention the reactions with tertiary amine to complete the full map of kinetics.

Response:

The CO₂ absorptions with primary and secondary amines (MEA and DEA) possess the similar reaction schemes with carbamate as product and involve Zwitterion mechanism [4].

For tertiary amines, the CO₂ absorption has the different reaction scheme with bicarbonate as product and involves 'Base Catalyzed hydration mechanism' [2].

Changes: Section 1, Introduction: Line 10, added:

“For the CO₂ reactions with tertiary amines, the product is bicarbonate (HCO₃⁻) and the mechanism is “Base Catalyzed hydration Mechanism”. [2] For the CO₂ reactions with primary and secondary amines, the product is carbamate (RR'N-COO⁻) and the mechanism is mostly Zwitterion mechanism [4, 6], except for Termolecular mechanism under special cases [17]. The focus of this study was based on the Zwitterion mechanism of carbamate formation with heterogeneous catalysis.”

Question 2. How to obtain activation energy Ea for reaction.

2. Theory

(2) In Section 2.2 of reaction (B2), the author mentioned the activation energy (Ea) of carbamate formation. The Ea of non-catalytic reaction can be calculated from rate constant from kinetic analysis. How to obtain the Ea of catalytic reaction?

Response:

The reviewer's suggestion is well taken, and the activation energy of catalytic reaction could be obtained with two possible methods.

The 1 st method is also calculated from rate constant from kinetic analysis. The reaction rates are tested under different temperatures and with the equation of $\ln k = E_a/RT$.

The 2 nd method involves molecular simulation. The E_a could be calculated with software simulation such as Gaussian 09. The E_a is the energy difference between the transition state and the ground state of reactants.

Changes: Section 2.2 Theory, Paragraph 3

The exact activation energy of (B2) was not tested, but it could be calculated with simulation or tested with further kinetic analysis with the equation of $\ln k = E_a/RT$.

Question 3. The application of B1-B4 to R3N

(3) Why these elementary steps B1-B4 unsuitable for tertiary amine? Please explain briefly.

Response:

The elementary steps B1-B4 are unsuitable because there is no carbamate generated within the $\text{CO}_2\text{-R}_3\text{N}$ reactions.

Similar to Question #1, the reason is also due to different reaction schemes, mechanisms, and products of $\text{CO}_2\text{-R}_3\text{N}$ reactions (bicarbonate rather than carbamate).

Table 2. The elementary reaction steps and rate law if rate determining of Fig 1.

No.	Elementary reaction steps of catalytic carbamate formation Eley-Rideal model ^a	Rate constant
B1	$\text{RNH}_2 + (\text{s}) \rightleftharpoons \text{RNH}_{2(\text{s})}$	k_1, k_{-1}
B2	$\text{CO}_2 (\text{g}) + \text{RNH}_{2(\text{s})} \rightleftharpoons \text{RNH}_2^+\text{-COO}^-_{(\text{s})}$ (Zwitterion)	k_2, k_{-2}
B3	$\text{H}_2\text{O} + \text{RNH}_2^+\text{-COO}^-_{(\text{s})} \rightarrow \text{RNH-COO}^-_{(\text{s})}$ (Carbamate) + H_3O^+	k_3, k_{-3}
B4	$\text{RNH-COO}^-_{(\text{s})} \rightleftharpoons \text{RNH-COO}^-_{(\text{s})}$ (Carbamate) + (s)	k_4, k_{-4}
RDS	Rate law if rate determining, with full and simplified formats	

From B1 to B4 in Table 2, these elementary steps involve carbamate (RNH-COO^-) as products. However, the main product is bicarbonate (HCO_3^-) for CO_2 reaction with tertiary amines ($\text{R}_1\text{R}_2\text{R}_3\text{N}$).

Changes: Section 2.2 Theory Last paragraph.

“These elementary steps were suitable for both primary and secondary amines (MEA, DEA, DIPA, MMEA etc., with unified form of $\text{RR}'\text{NH}$) with “carbamate ($\text{RR}'\text{N-COO}^-$)” as product. However, they were unsuitable for tertiary amine (R_3N). The $\text{CO}_2\text{-R}_3\text{N}$ reactions involved different reaction schemes, mechanisms and products. The main product was bicarbonate (HCO_3^-) for CO_2 reaction with tertiary amines ($\text{R}_1\text{R}_2\text{R}_3\text{N}$) so that B1-B4 were unsuitable for it.”

Question 4. The reaction process on the solid surface

3. Result and Discussion

(4) Section 4.3 First paragraph, “The solid surface had abundant empty active sites with very few carbamate adsorbed.” Please provide a detailed explanation of this sentence.

Response:

The reviewers' suggestion is taken, and the overall adsorption process were briefly introduced.

Changes: Section 4.3, second paragraph

“The overall CO₂ absorption process with existence of solid alkaline was briefly explained from B1 to B4 in Table 2. The amine adsorbed onto the surface from liquid phase firstly, with relatively slow rate (B1). Then CO₂ reacted with MEA (RNH₂) with N-C bond formation (B2). The rate was instantaneous and enhanced with heterogeneous catalysis. The Zwitterion released proton to H₂O or other base to generate carbamate (RNH-COO⁻) and the exothermic reaction released heat (B3). The released heat facilitated diffusion and drove desorption of carbamate back to the aqueous phase (B4). The carbamate finally desorbed the surface due to the exothermic reaction. [24] Mostly the product desorbed the solid surface due to the heat release, and there was little carbamate remained.”

Question 5. The future work

(5) Last paragraph, “This Eley-Rideal model still awaits much further analyses with updated experimental apparatus”, what types of experiments did you wish to conduct as the next task?

Response:

The reviewer raised a good question. This model requires massive data for improvements and verifications. Two types of studies could be carried out next:

- (1) The in situ study of a new solid base catalyst, KMgO/CNTs (carbon nanotubes), which was reported [30] recently to be useful for CO₂-MEA absorption. This work proves the applicability of this model to CO₂-MEA with various solid base catalysts.
- (2) The molecular simulation can be performed for CO₂-MEA reactions with existence of CaCO₃ with Gaussian 09. The reactant, transition state and product can be simulated and the energy can be calculated to address the activation energy E_a of the catalytic absorption. After compared with E_a of non-catalytic absorption, the effectiveness of catalysts can be measured quantitatively.

Changes: Last paragraph of Section 4.3

The molecular simulation needed to be calculated of catalytic CO₂-MEA reactions with existence of CaCO₃ with Density Function Theory (DFT) to discover E_a of the catalytic absorption. The kinetic

analysis of another solid base “KMgO/CNT (carbon nano-tubes)” with ER model could be performed since this material was reported to be effective for CO₂-MEA absorption recently. [30]

Question 6. The RDS among B1 – B4 for DEA

4. Conclusions

(6) For catalytic CO₂-DEA reaction with this model, is RDS still B1?

Response:

The reviewer has raised a very good question. The answer remains unknown, which requires numerous studies with intensive literature review and detailed investigations.

The B1-B4 are suitable for most primary and secondary amines such as MEA and DEA. However, each amine has different rate constant, and the verification of RDS requires both theoretical analysis and experimental tests.

For MEA, the case is easier and straight forward since the CO₂ reaction with MEA is instantaneous and the reaction order is 1. For DEA, the case is special, because the reaction order is between 1-2 of [DEA]. We can only make such statement now. The RDS should belong to any step of B1-B4, but it requires intensive literature studies and experimental results based on highly accurate kinetic studies with specialized reactor to verify the equations properly.

Changes: Conclusions Line 4,

For DEA, the rate-determining step was among B1-B4, but it required intensive literature studies and experimental results based on highly accurate kinetic studies with specialized reactor to verify the equations properly.

Question 7. The dominant effect of increased reaction rates

(7) Last sentence, the increased reaction rates is result in either mass transfer or reduced activation energy. Which part is dominant and How to distinguish these effects?

Response:

The reviewer has raised a good question. For the catalyst, there are several steps during catalytic process. They are external diffusion, internal diffusion, adsorption, surface reactions, internal diffusion, external diffusion and desorption.

Both parts are important, for the catalyst not only enhances the mass transfer with active sites on the surface but also reduces the activation energy (E_a) to accelerate reactions. However, based on the Arrhenius equation, the reduction of E_a is more important.

There are different experiments to test these effects. The effect of mass transfer study can be conducted in a CO₂ absorber to test k_{Gav} . The activation energy can be calculated with molecular simulation first and then calculated based on Arrhenius equation with experimental data of kinetic analyses.

Changes: Conclusion, Last sentence

“The reduction of activation energy E_a was more important in accelerating reaction rates. The effect of mass transfer could be calculated from k_{Gav} on the basis of experiments, and E_a could be calculated from either molecular simulation or Arrhenius equation with the aid of experimental data of kinetics analyses.”

Appendix B

Respond to Editor:

1. Ethics statement was completed, according to the terms of the Creative Commons Attribution License <http://creativecommons.org/licenses/by/4.0/>. The PDF was also uploaded.
2. Acknowledgements to Yunlong Zhou, who contributed to the study but did not meet the authorship criteria.

Respond to Reviewer comments to Author:

Reviewer: 2

Comments to the Author(s)

The author did responded to all the questions, and provided reasonable explanations. The changes were appropriate and helpful for the readers to understand the manuscript.

Response: Thanks for the reviewer.

Reviewer: 1

Comments to the Author(s)

The authors answer most of the questions from reviewers. There are two minor but necessary modifications needed:

1. In the described mechanisms, the arrows showed the direction of the molecule (such as CO₂) attacking the adsorbed amines. This counters the common convention that the arrows indicate the flow direction of the electrons. Using this example, the lone pair electrons from the N of the amine should attack the carbon of the CO₂ to form the C-N bond. So under the usual convention, the direction of the arrow should go from the amine to the carbon of the CO₂. The authors should use the typical convention instead.

Response:

The reviewer's suggestion was very good, the reaction is the long pair electrons on the N atom of amine to attach to C atom of CO₂.

Changes: The new Figure 1 was generated

2. There are many typos, random capitalization of words and missed subscripts in the manuscript and the references. There are too many of them to be listed here. These to be corrected.

Response:

The reviewer's suggestion was taken, and we proofread the whole manuscript thoroughly.